# TideGS: Scalable Training of Over One Billion 3D Gaussian Splatting Primitives via Out-of-Core Optimization

Chonghao Zhong[1]   Linfeng Shi[1]   Hua Chen[2]   Tiecheng Sun[2]   Hao Zhao[3][4]   Binhang Yuan[*][1]   Chaojian Li[*][1]

## Abstract

Training 3D Gaussian Splatting (3DGS) at billion-primitive scale is fundamentally memory-bound: each Gaussian primitive carries a large attribute vector, and the aggregate parameter table quickly exceeds GPU capacity, limiting prior systems to tens of millions of Gaussians on commodity single-GPU hardware. We observe that 3DGS training is inherently sparse and trajectory-conditioned: each iteration activates only the Gaussians visible from the current camera batch, so GPU memory can serve as a working-set cache rather than a persistent parameter store. Building on this insight, we introduce **TideGS**, an out-of-core training framework that manages parameters across an SSD–CPU–GPU hierarchy via three synergistic techniques: block-virtualized geometry for SSD-aligned spatial locality, a hierarchical asynchronous pipeline to overlap I/O with computation, and trajectory-adaptive differential streaming that transfers only incremental working-set deltas between iterations. Experiments show that TideGS enables training with **over one billion Gaussians** on a single 24 GB GPU while achieving the best reconstruction quality among evaluated single-GPU baselines on large-scale scenes, scaling beyond prior out-of-core baselines (e.g., ~100M Gaussians) and standard in-memory training (e.g., ~11M Gaussians). Project page.

## 1. Introduction

3D Gaussian Splatting (3DGS) (Kerbl et al., 2023) has emerged as a strong representation for novel view synthesis, combining explicit scene primitives with an efficient

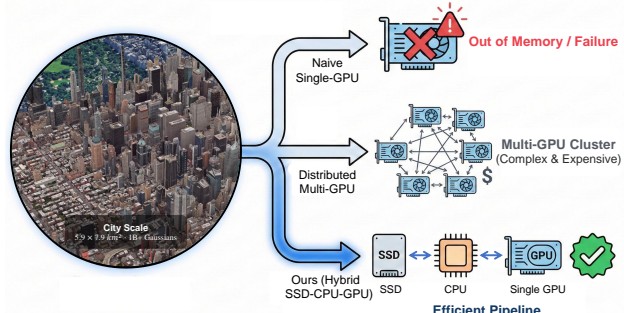

*Figure 1.* TideGS enables city-scale 3DGS training on a single GPU by virtualizing the Gaussian parameter table across the SSD–CPU–GPU hierarchy and materializing only the trajectory-activated working set in VRAM.

rasterization-based rendering pipeline (Zhang et al., 2024; Hanson et al., 2025; Lan et al., 2025; Ren et al., 2026; Liao et al., 2025; Gui et al., 2024; Xu, 2024; Tian et al., 2025; Fang & Wang, 2024; Mallick et al., 2024; Feng et al., 2025; Wang et al., 2025). By representing a scene as a collection of anisotropic Gaussians with learned appearance parameters, 3DGS achieves high-fidelity novel view synthesis while supporting real-time rendering. This explicit representation also changes the scaling bottleneck: compared with implicit neural representations such as NeRF (Mildenhall et al., 2022; Müller et al., 2022; Yuan & Zhao, 2024; Liu et al., 2024), 3DGS shifts much of the model capacity into a large primitive table, making training increasingly memory-bound as scene scale grows.

Despite this progress, scaling 3DGS training to large scenes remains fundamentally constrained by memory. Each Gaussian is parameterized by 59 floating-point values spanning geometric attributes and spherical harmonic coefficients (Kerbl et al., 2023). During training, parameters, gradients, and optimizer states (e.g., Adam moments) require multiple copies of these values. Consequently, a scene with 100 million Gaussians demands nearly 90 GB of memory, exceeding a typical 24 GB single-GPU memory budget and even stressing high-end datacenter accelerators. In practice, model capacity quickly saturates: on a 24 GB GPU, vanilla 3DGS (Kerbl et al., 2023) typically reaches only the ~11M-Gaussian regime, and optimized host-offloading pipelines (Zhao et al., 2026) remain around ~100M Gaussians. Meanwhile, prior work (Li et al., 2024; Zhao et al., 2025; Lee et al., 2026) suggests that increasing the number

---

[1]Hong Kong University of Science and Technology [2]Great Wall Motor [3]Tsinghua University [4]Beijing Academy of Artificial Intelligence. Correspondence to: Binhang Yuan <biyuan@ust.hk>, Chaojian Li <chaojian@ust.hk>.

*Proceedings of the 43$^{rd}$ International Conference on Machine Learning*, Seoul, South Korea. PMLR 306, 2026. Copyright 2026 by the author(s).

of Gaussians can improve rendering fidelity, especially for large-scale environments such as aerial captures and urban street scenes. Multi-GPU systems can scale by aggregating device memory (Zhao et al., 2025; Li et al., 2024), but they introduce substantial infrastructure cost and engineering complexity. These trends make single-GPU scalability a central bottleneck for accessible large-scale 3DGS training.

The key opportunity is that 3DGS optimization does not access the full parameter table at every step. For a given camera batch, only visible Gaussians participate in rasterization and receive non-zero gradients, while most primitives remain inactive. This visibility-induced sparsity resembles sparse embedding-table training (Wilkening et al., 2021) and motivates treating VRAM as a high-bandwidth working-set cache rather than a persistent parameter store. Prior host-offloading methods (Lee et al., 2026; Zhao et al., 2026) exploit part of this structure but still keep key geometry GPU-resident, effectively capping single-GPU scalability near the ~100M-Gaussian regime. Scaling beyond this point requires extending the hierarchy to SSD storage, where much lower bandwidth and higher latency make naive offloading impractical.

Building on this cache-centric view, we introduce **TideGS**, an out-of-core training framework that manages 3DGS parameters across an SSD–CPU–GPU hierarchy. TideGS combines three techniques: (i) *block-virtualized geometry*, which packs spatially coherent Gaussians into SSD-aligned blocks; (ii) a *hierarchical asynchronous pipeline*, which overlaps SSD reads, host–device transfers, write-back, and GPU rendering/backpropagation; and (iii) *trajectory-adaptive differential streaming*, which retains overlapping working sets across nearby views and transfers only incremental block deltas. Together, these designs make SSD-tier out-of-core training practical by bounding communication to visible working-set changes while preserving the standard 3DGS forward/backward semantics on the resident primitives.

Our experiments show that TideGS trains scenes with over one billion Gaussians on a single 24 GB GPU while achieving high reconstruction quality on city-scale scenes. At in-memory-feasible scales, TideGS preserves Native 3DGS quality and incurs only modest overhead ($<15\%$) over GPU-resident training; in the out-of-core regime, it remains throughput-competitive while scaling an order of magnitude beyond prior single-GPU methods. These results establish out-of-core optimization as a practical path toward scalable and accessible 3DGS training.

## 2. Preliminaries

**Per-Gaussian parameter table.** In standard 3DGS (Kerbl et al., 2023), each Gaussian primitive $i$ carries a learnable

parameter vector $\theta_i \in \mathbb{R}^D$ that encodes geometry and appearance; under the standard degree-3 SH parameterization used throughout this paper, $D=59$. For $N$ Gaussians, these vectors form a dense parameter table $\Theta \in \mathbb{R}^{N \times D}$. Training also maintains gradients and optimizer states such as Adam moments, so the total state size grows linearly with $N$ but with a large constant factor. This parameter-table view makes the VRAM bottleneck explicit: scaling scene capacity requires managing not only the Gaussian attributes but also their training states.

**Visibility-induced sparse updates.** Although the full table may be large, each training iteration touches only the Gaussians that contribute to the current camera batch. For a batch $\mathcal{B}_t$ at iteration $t$, let $\mathcal{I}_t \subseteq \{1, \ldots, N\}$ denote the union of Gaussian indices that are visible after rasterization and receive non-zero gradients. In large scenes, this active set is typically much smaller than the full model, i.e., $|\mathcal{I}_t| \ll N$. Moreover, when batches follow nearby viewpoints along a smooth camera trajectory, the active sets of adjacent iterations often overlap substantially. This creates both sparsity (small active sets) and temporal locality (similar active sets across adjacent iterations).

**Block-level working sets.** Out-of-core storage cannot efficiently fetch individual Gaussians one by one, so TideGS uses blocks as the transfer and cache unit. For $K$ blocks indexed by $k \in \{0, \ldots, K-1\}$, let $\text{Block}(k) \subseteq \{1, \ldots, N\}$ be the Gaussian indices assigned to block $k$. At iteration $t$, the block-level working set $\mathcal{K}_t$ contains the blocks that conservatively cover the Gaussian-level active set $\mathcal{I}_t$. The subsequent method therefore separates two granularities: block-level staging and caching are performed over $\mathcal{K}_t$, while fine-grained rendering and gradient updates are still applied to Gaussians in $\mathcal{I}_t$.

**Out-of-core training.** When the full training state exceeds GPU VRAM, offloading keeps most state on a slower tier such as CPU DRAM or SSD and materializes only the current working set on GPU (Ren et al., 2021). Practical throughput then depends on two properties: the staged working set must remain small through sparse, locality-preserving access, and data movement must overlap with rendering/backpropagation to hide transfer latency. These requirements become stricter at the SSD tier, where bandwidth is lower and latency is higher than GPU or CPU memory. TideGS is designed around these constraints by turning 3DGS visibility sparsity and trajectory locality into block-level out-of-core execution.

## 3. Method

We present **TideGS**, an out-of-core training framework that enables billion-scale 3D Gaussian Splatting (3DGS) on a

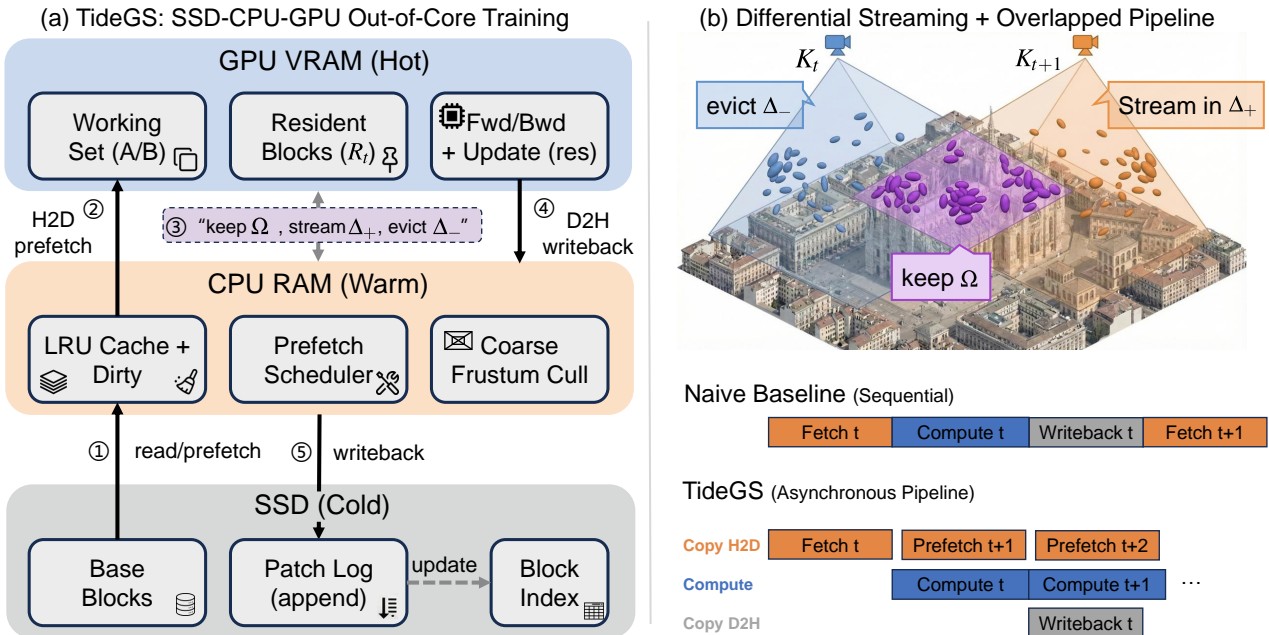

*Figure 2.* **TideGS pipeline. (a) Out-of-core hierarchy.** Full-scene blocks reside on SSD; CPU RAM provides warm caching and prefetch scheduling; GPU VRAM caches resident blocks for rendering/backpropagation. **(b) Trajectory-adaptive differential streaming.** Consecutive block working sets $\mathcal{K}_t$ and $\mathcal{K}_{t+1}$ guide reuse, while TideGS selects capacity-bounded resident sets $\mathcal{R}_t$ and $\mathcal{R}_{t+1}$, keeps their overlap, streams only $\mathcal{S}_t^+ = \mathcal{R}_{t+1} \setminus \mathcal{R}_t$, and evicts $\mathcal{S}_t^- = \mathcal{R}_t \setminus \mathcal{R}_{t+1}$ while overlapping SSD/PCIe transfers with GPU compute. Circled numbers indicate cross-tier dataflow steps.

single 24 GB GPU using commodity CPU memory and SSD storage. As illustrated in Fig. 2, TideGS treats GPU VRAM as a high-bandwidth working-set cache: at iteration $t$, only the blocks needed by the current camera batch are materialized in VRAM, while the full parameter table remains SSD-resident and is accessed through a coordinated SSD–CPU–GPU hierarchy. TideGS makes SSD-tier out-of-core training practical through block-level parameter virtualization, asynchronous cross-tier pipelining, and trajectory-adaptive reuse across iterations.

### 3.1. Problem Setting: Sparse, View-Dependent Working Sets

3DGS training exhibits strong *visibility sparsity*: for a camera batch $\mathcal{B}_t$, only a small subset of Gaussians receives non-zero gradients. As defined in Sec. 2, we distinguish two granularities: $\mathcal{I}_t$ denotes the Gaussian-level active set, while $\mathcal{K}_t$ denotes its conservative block-level cover used for staging and caching. This sparsity is also observed empirically in prior systems: CLM reports that on the MatrixCity BigCity/Aerial subset (Li et al., 2023), a single view accesses only $0.39\%$ of Gaussians on average (up to $1.06\%$ in the worst case) (Zhao et al., 2026). Moreover, under smooth camera motion, consecutive iterations tend to access highly overlapping block working sets, so the incremental change in the working set is often much smaller than the working set itself. In an out-of-core setting, the system should therefore

make cross-tier traffic scale with the visible block working set, and especially with its incremental change over time, rather than with the full model size.

### 3.2. System Overview

Fig. 2 summarizes the resulting training loop. At iteration $t$, TideGS identifies the block working set $\mathcal{K}_t$, maintains a VRAM-resident set $\mathcal{R}_t$ under capacity $C$, stages only incoming blocks, and asynchronously writes back evicted dirty blocks. Each iteration has four coordinated stages: **Stage 1: Identify working set** by computing $\mathcal{K}_t$ with lightweight CPU-side block visibility tests (Sec. 3.3); **Stage 2: Prefetch & materialize** needed blocks through the CPU cache and an asynchronous host-to-device (H2D) stream (Sec. 3.4); **Stage 3: Render & backprop** with the standard 3DGS forward/backward pass on resident blocks $\mathcal{R}_t$; and **Stage 4: Evict & write back** cold blocks, propagating dirty evictions through the CPU cache to SSD patch segments (Sec. 3.4). Stages (1)/(2)/(4) overlap with (3) whenever possible, amortizing SSD/PCIe latency with GPU computation.

### 3.3. Block Virtualization and Two-Stage Visibility Filtering

TideGS converts per-iteration Gaussian visibility into a block-level working set that can be fetched and cached efficiently in an out-of-core setting. The key idea is to

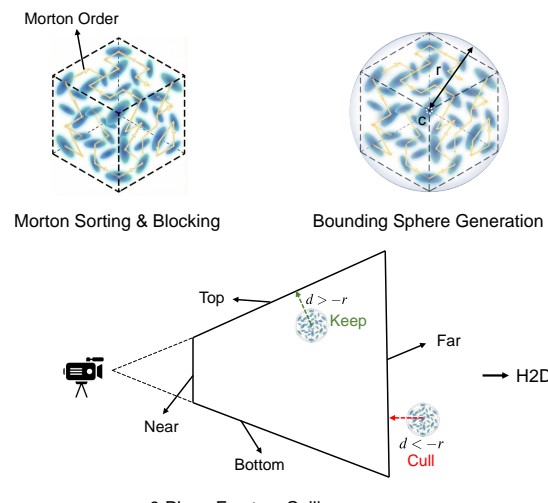

*Figure 3.* **Block virtualization and two-stage visibility filtering.** We (top-left) Morton-sort Gaussians and partition them into SSD-aligned blocks to preserve spatial locality, (top-right) summarize each block with a bounding sphere, and (bottom) perform CPU-side 6-plane frustum tests to select visible blocks for H2D staging. Fine-grained Gaussian-level filtering is then executed on GPU within the retained blocks.

(i) pack per-Gaussian parameters into SSD-aligned spatial blocks and (ii) run a conservative CPU-side block visibility test before data movement, while preserving exact 3DGS semantics through GPU-side fine filtering.

**Unified parameter layout.** We use a unified logical layout $\Theta \in \mathbb{R}^{N \times D}$ for all learnable per-Gaussian attributes, with $D=59$ under the standard degree-3 SH parameterization used throughout this paper. Physically, $\Theta$ is stored out-of-core as contiguous block records on SSD, while CPU and GPU memory materialize only cached or resident blocks.

**SSD-aligned blocks with spatial locality.** In the logical layout $\Theta$, each block corresponds to a contiguous row range of Gaussian parameters:

$$\text{Block}(k) := \Theta[kB : (k+1)B], \quad k \in \{0, \dots, K-1\}. \tag{1}$$

Here $K = \lceil N/B \rceil$, and the final range is truncated at $N$ when $N$ is not divisible by $B$. We set $B=4096$. With fp32 parameters and $D=59$, each full block has a parameter payload of $4096 \times 59 \times 4$ bytes, i.e., 236 contiguous 4 KB pages (about 944 KiB). This aligns the dominant block records with common filesystem/page-cache granularities and improves the efficiency of buffered SSD reads/writes. To improve locality-aware reuse under camera motion, we Morton-sort Gaussians by the codes of their centers before blocking (Fig. 3, top-left), so spatially nearby Gaussians map to nearby indices and thus nearby blocks. Intuitively, spatially compact blocks yield tighter bounding spheres, improving the precision of CPU-side frustum culling. After

initialization, the owner block of each Gaussian is fixed: center updates during training change the Gaussian's position but do not migrate or duplicate the primitive across blocks. We conservatively refresh each affected block bound as centers move, so neighboring block bounds may overlap, but each Gaussian remains uniquely owned and is rasterized exactly once. Thus, block virtualization preserves the standard 3DGS rendering semantics while allowing block-level storage and streaming.

**Level 1 (CPU): coarse block visibility via frustum culling.** Fine-grained visibility tests over all $N$ Gaussians are unnecessary and expensive at large scale, and more importantly they would force SSD/PCIe traffic to scale with $N$. TideGS therefore first computes the active block set on CPU before any GPU transfers. Each block $k$ is summarized by a coarse bounding volume; we use a bounding sphere with center $\mathbf{c}_k$ and radius $r_k$ (Fig. 3, top-right). Given a camera batch $\mathcal{B}_t$, we apply a standard 6-plane frustum test to these spheres and keep only intersecting blocks:

$$\mathcal{K}_t = \mathcal{K}(\mathcal{B}_t) = \bigcup_{c \in \mathcal{B}_t} \{k \mid \text{visible}(k, c)\}. \tag{2}$$

Equivalently, for each frustum plane, we cull a sphere if its signed distance to the plane satisfies $d < -r_k$ (Fig. 3, bottom). This coarse filtering ensures that subsequent SSD/PCIe transfers scale with the selected block working set, rather than with the full model size $N$.

**Level 2 (GPU): fine filtering and exact rendering within active blocks.** After residency selection materializes the selected resident blocks $\mathcal{R}_t$ in VRAM, TideGS runs the standard 3DGS projection/rasterization pipeline on the resident visible blocks. Gaussian-level culling and rasterization determine the final contributing set $\mathcal{I}_t \subseteq \bigcup_{k \in (\mathcal{R}_t \cap \mathcal{K}_t)} \text{Block}(k)$. Only Gaussians in $\mathcal{I}_t$ participate in forward/backward and receive non-zero gradients. Level 1 is conservative (it may admit extra blocks), and Level 2 applies the exact 3DGS pipeline within the resident visible blocks; therefore, the rendering/backpropagation kernels and per-Gaussian update semantics are unchanged.

### 3.4. Out-of-Core Engine: SSD Storage, CPU Tiered Cache, and Asynchronous Execution

TideGS maintains the full block array on SSD while keeping the GPU compute path throughput-competitive. The out-of-core engine must (i) avoid random SSD writes under frequent parameter updates, (ii) exploit CPU DRAM as a warm cache between SSD and VRAM, and (iii) overlap SSD/PCIe transfers with GPU rendering/backpropagation.

**Log-structured SSD storage.** TideGS organizes SSD storage as log-structured append-only segments. The ini-

tial model is written once as an immutable base segment. During training, updated blocks are written sequentially into patch segments rather than overwriting existing block locations in place. Each patch segment contains a batch of updated block versions produced by a cache flush. We maintain a per-block pointer to the latest version:

$$\text{Index}[k] = (\text{file\_id}, \text{offset}, \text{size}, \text{version}). \quad (3)$$

Here $\text{file\_id} = 0$ denotes the base segment and later file IDs denote patch segments. Reads consult $\text{Index}$ to materialize the newest version of each block. By avoiding in-place overwrites, the write path becomes sequential and achieves high sustained throughput. Optional compaction can merge patch segments into a new base segment, but this is outside the training critical path.

**CPU cache with write-back and dirty tracking.** CPU DRAM serves as a warm cache between SSD and GPU. We maintain an LRU cache over blocks together with a per-block dirty bit. A block is marked *dirty* only when its parameters have been updated by GPU-side training. The LRU policy is updated on each access and is independent of the dirty bit: frequently reused dirty blocks may remain resident in CPU memory and are not immediately persisted to SSD. Dirty blocks are flushed to SSD patch segments when they are evicted under CPU memory pressure, or at explicit consistency barriers such as checkpointing and shutdown.

**Two-step eviction and write-back: VRAM → CPU → SSD.** To decouple GPU residency from SSD write latency, TideGS employs a two-step write-back path. When a block is evicted from VRAM to make room for incoming blocks, it is transferred via D2H and inserted into the CPU cache. Clean blocks are inserted as clean entries, while dirty blocks are inserted as dirty entries. During normal training, when a dirty block is later evicted from the CPU cache, we asynchronously flush it to SSD patch segments, append a new version, and update $\text{Index}[k]$ to point to the latest location. On re-admission, blocks are always fetched via $\text{Index}[k]$, so the GPU always materializes the most recent version.

**Hierarchical asynchronous execution.** A naive out-of-core loop would stall on SSD reads and PCIe transfers. TideGS overlaps four operations to avoid stalls: (i) SSD read/prefetch into the CPU cache, (ii) H2D transfer to materialize the incoming blocks in VRAM, (iii) GPU compute on the resident set, and (iv) D2H transfer of evicted blocks plus asynchronous SSD flush from the CPU cache. Implementation-wise, TideGS runs SSD read/prefetch/flush in dedicated I/O threads, manages caching and dirty tracking on the CPU, and uses separate CUDA streams for GPU compute and copies. With double-buffered GPU block buffers,

TideGS transfers the next iteration's incoming blocks while computing the current iteration, matching Fig. 2(b).

### 3.5. Tide: Trajectory-Adaptive Differential Streaming

Even after coarse block-level culling, materializing the full visible block union $\mathcal{K}_t$ in VRAM at every iteration is wasteful under smooth camera motion, because consecutive batches often access highly overlapping block sets. TideGS therefore reuses resident blocks across iterations and transfers only incoming resident deltas. We use a clustered TSP-ordered (no-shuffle) camera sequence to increase overlap between consecutive block working sets; convergence is discussed in Appendix A.2.

**Residency scoring.** When VRAM is capacity-limited, TideGS maintains a capacity-bounded resident set $\mathcal{R}_t$ rather than materializing the full visible block set $\mathcal{K}_t$. For the next iteration, we form a candidate pool $\mathcal{C}_t = \mathcal{R}_t \cup \mathcal{K}_{t+1}$ and score each candidate block by combining *next-step usefulness* and *recency*:

$$s(k) = \lambda \cdot \mathbf{1}[k \in \mathcal{K}_{t+1}] + (1 - \lambda) \cdot \text{Recency}(k). \quad (4)$$

Here $\text{Recency}(k)$ is an LRU-style recency score updated on each access (reset on access and aged otherwise), and $\lambda \in [0, 1]$ controls the trade-off between prioritizing the next working set and retaining recently used blocks. When $|\mathcal{K}_{t+1}| > C$, a pure global Top-$C$ selection may leave some views in a mini-batch under-covered. We therefore use a camera-balanced Top-$C$ policy: a small quota of resident slots is first assigned to cover visible blocks from each camera in the next batch, and the remaining slots are filled by the global score $s(k)$ over $\mathcal{C}_t$. This produces the next resident set $\mathcal{R}_{t+1}$ under budget $C$.

**Set-difference streaming.** Given the current and next resident sets, TideGS keeps the resident overlap and transfers only the delta:

$$\Omega_t^R = \mathcal{R}_t \cap \mathcal{R}_{t+1}, \quad \mathcal{S}_t^+ = \mathcal{R}_{t+1} \setminus \mathcal{R}_t, \quad \mathcal{S}_t^- = \mathcal{R}_t \setminus \mathcal{R}_{t+1}. \quad (5)$$

Thus, TideGS retains $\Omega_t^R$ in VRAM, streams only $\mathcal{S}_t^+$, and evicts $\mathcal{S}_t^-$, so PCIe volume scales with *resident-set change* rather than the full model size. Algorithm 1 summarizes the resulting camera-balanced residency selection and set-difference transfer procedure.

**GPU-side training on the resident set.** TideGS executes rendering and backpropagation on GPU using the capacity-bounded resident set $\mathcal{R}_t$. The coarse visible block set $\mathcal{K}_t$ defines the candidate working set for the current batch, while $\mathcal{R}_t$ is the set actually materialized in VRAM after residency selection and reuse under budget $C$. Resident blocks that participate in the current forward/backward pass and receive

gradient updates are marked dirty and follow the write-back policy.

**Lazy write-back through the CPU cache.**    To avoid frequent small SSD writes, TideGS decouples eviction from VRAM and persistence on SSD. When a dirty block is evicted from VRAM, it is staged to CPU and inserted into the CPU cache as dirty; during normal training, it is appended to SSD patch segments only when it is later evicted from the CPU cache. Explicit consistency barriers may also flush dirty CPU-cache entries as needed. This design amortizes write-back and turns frequent block updates into batched sequential appends on the SSD write path, avoiding random in-place overwrites.

---

**Algorithm 1** Tide residency selection and differential streaming

---

**Require:** Current visible block set $\mathcal{K}_t$, next camera-wise block sets $\{\mathcal{K}_{t+1}^{(j)}\}_{j=1}^J$
**Require:** Current resident set $\mathcal{R}_t$, block capacity $C$
**Require:** Recency score $\mathrm{Recency}(\cdot)$, mixing weight $\lambda \in [0, 1]$
**Ensure:** Next resident set $\mathcal{R}_{t+1}$, stream-in blocks $\mathcal{S}_t^+$, evict blocks $\mathcal{S}_t^-$
1: $\mathcal{K}_{t+1} \leftarrow \bigcup_{j=1}^J \mathcal{K}_{t+1}^{(j)}$
2: Update $\mathrm{Recency}(\cdot)$ from current accessed resident blocks in $\mathcal{R}_t \cap \mathcal{K}_t$
3: $\mathcal{C}_t \leftarrow \mathcal{R}_t \cup \mathcal{K}_{t+1}$
4: **for all** $k \in \mathcal{C}_t$ **do**
5:     $s(k) \leftarrow \lambda \cdot \mathbf{1}[k \in \mathcal{K}_{t+1}] + (1 - \lambda) \cdot \mathrm{Recency}(k)$
6: **end for**
7: $\mathcal{R}_{t+1} \leftarrow \mathrm{CameraBalancedTopC}\big(\{\mathcal{K}_{t+1}^{(j)}\}, \mathcal{C}_t, s, C\big)$
8: $\Omega_t^R \leftarrow \mathcal{R}_t \cap \mathcal{R}_{t+1}$
9: $\mathcal{S}_t^+ \leftarrow \mathcal{R}_{t+1} \setminus \mathcal{R}_t$;  $\mathcal{S}_t^- \leftarrow \mathcal{R}_t \setminus \mathcal{R}_{t+1}$
10: **return** $\mathcal{R}_{t+1}, \mathcal{S}_t^+, \mathcal{S}_t^-$

---

**Optimizer state placement.**    TideGS keeps the full model out-of-core; only the resident working set is materialized in VRAM. By default, optimizer states (e.g., Adam moments) are instantiated only for resident blocks and discarded upon eviction (cold restart on re-admission). This design trades optimizer-state persistence for lower cross-tier traffic and a smaller VRAM footprint. Under trajectory-adaptive execution, hot blocks can remain resident across consecutive nearby views, so their optimizer states are preserved while they stay hot, whereas re-admitted blocks cold-start their moments. Appendix A.5 reports the resulting churn statistics, including the cold-restarted update ratio, eviction/re-admission rates, and mean resident streaks.

## 4. Experiments

We design our evaluation to answer four research questions: (1) **Scalability:** What is the maximum trainable scale on a single 24 GB GPU (a representative single-GPU memory budget), and what bottleneck limits each baseline? (2) **Over-**

**head:** Does TideGS introduce measurable overhead relative to in-memory training at scales that fit in VRAM? (3) **Efficiency:** Under large-scale training, how does TideGS compare to host-offloading baselines in throughput and cross-tier traffic? (4) **Quality:** Does scaling to more Gaussian primitives improve reconstruction quality on city-scale scenes? We answer these questions using standard benchmarks and a city-scale dataset, together with detailed system measurements that separate reconstruction quality, throughput, data movement, and resource usage.

### 4.1. Experimental Setup

**Setup.**    Unless otherwise noted, experiments run on one NVIDIA RTX A5000 GPU (24 GB VRAM), an AMD EPYC 7532 CPU, 256 GB DDR4 memory, and a Samsung PM9A3 NVMe SSD with 3.3 GB/s measured I/O speed. We evaluate Mip-NeRF 360 (Barron et al., 2022) for in-memory quality/overhead and MatrixCity BigCity/Aerial (Li et al., 2023) for city-scale out-of-core scalability, traffic, and quality. Baselines include Native 3DGS (Kerbl et al., 2023), a ZeRO-Offload-inspired (Ren et al., 2021) Naive Offload baseline, and CLM (Zhao et al., 2026); multi-GPU systems are discussed in Appendix A.6. TideGS uses block size $B = 4096$ and Adam with the 3DGS learning-rate schedule. We use batch size $|\mathcal{B}_t| = 4$ for 30k Mip-NeRF 360 iterations. For MatrixCity, $|\mathcal{B}_t| = 64$ with a 16 GB CPU cache is used for throughput and traffic measurements by default, while $|\mathcal{B}_t| = 16$ with a 32 GB CPU cache is used for memory-pressure-sensitive runs. Additional measurement controls and preprocessing details are provided in Appendix A.1.

*Table 1.* **Scalability frontier: the "VRAM wall" and out-of-core scaling.** We report the maximum trainable scale ($N_{\max}$) on a single 24 GB GPU. **Native 3DGS** is limited by full training state in VRAM. **Naive Offload** remains parameter-limited due to GPU-resident per-iteration parameters (dominated by SH). **CLM** (Zhao et al., 2026) offloads SH but is ultimately bounded by VRAM-hungry rasterization buffers at large $N$. **TideGS** makes VRAM usage depend on the capacity-bounded resident set $|\mathcal{R}_t|$ and Gaussian active set $|\mathcal{I}_t|$, enabling billion-scale training with capacity primarily bounded by storage.

| Method | VRAM Scaling | Limiting Factor | Max $N$ |
|---|---|---|---|
| Native 3DGS (Kerbl et al., 2023) | $O(N)$ | Full state (VRAM) | $\sim$11.5M |
| Naive Offload | $O(N)$ | Parameters (SH) | $\sim$50M |
| CLM (Zhao et al., 2026) | $O(N)$ | Rasterization (VRAM) | $\sim$105M |
| **TideGS (Ours)** | $\mathbf{O(|\mathcal{R}_t|) / O(|\mathcal{I}_t|)}$ | Resident set / Storage | $> \mathbf{1B}$ |

### 4.2. Main Results

#### 4.2.1. SCALABILITY AND THE VRAM WALL

**Baselines and their limiting bottlenecks.**    As shown in Tab. 1, Native 3DGS (in-memory) fails at $\sim$**11.5M** Gaussians (measured), as all training states (parameters, gradients, optimizer states) must reside in VRAM. Naive Offload (parameter-limited) cannot scale far beyond tens of millions of Gaussians: although optimizer states are offloaded

to CPU memory, the per-iteration parameters required by rasterization must still reside in (or be repeatedly staged to) VRAM for rasterization/backprop, so the VRAM footprint still grows with the model size and does not decouple from $N$. For the Naive Offload baseline, profiling a 25M-Gaussian model shows 12.13 GB of GPU memory consumption for GPU-resident per-iteration parameters. A linear extrapolation therefore suggests an upper bound on the order of $\sim$**50M** Gaussians, often lower in practice due to additional rasterization buffers and allocator overhead. CLM reaches the $\sim$**100M** regime but becomes infeasible when pushed near $\sim$**105M** Gaussians: while CLM offloads the high-dimensional appearance attributes (spherical harmonics) and reduces parameter residency, the bottleneck shifts to VRAM-intensive rasterization buffers at large $N$. In our stress test near this scale, the global radix sort and auxiliary buffers exceed available VRAM (consuming >22 GB), triggering "Rasterization OOM".

**TideGS: shifting the limiting factor.** TideGS decouples VRAM usage from the total scene size $N$ by keeping the full parameter table out-of-core and materializing only the per-iteration working set. Accordingly, GPU memory scales with the capacity-bounded resident blocks and active Gaussians, i.e., $O(|\mathcal{R}_t|)$ (or $O(|\mathcal{I}_t|)$ after fine filtering), rather than $O(N)$. On large roaming scenes, we empirically observe $|\mathcal{K}_t| \ll N$, which keeps the candidate working set sparse relative to the full model, while the explicit budget on $|\mathcal{R}_t|$ keeps the VRAM footprint approximately stable as $N$ increases. As summarized in Tab. 1, TideGS is the only evaluated single-GPU method that scales beyond the VRAM wall, shifting the limiting factor from GPU memory to out-of-core storage capacity and bandwidth.

### 4.2.2. OVERHEAD IN THE IN-MEMORY REGIME

**Setup.** We evaluate on Mip-NeRF 360 (Barron et al., 2022) in an in-memory regime where Native 3DGS fits in GPU memory and TideGS can serve virtualized blocks from memory after warm-up. This setting isolates the software overhead of block virtualization and scheduling rather than physical SSD I/O. Results are averaged over the evaluated Mip-NeRF 360 scenes. We compare two TideGS scheduling variants: Shuffle randomizes the per-iteration camera order, while Trajectory uses the clustered TSP-ordered camera sequence described in Sec. 3.5 to preserve view locality and enable differential streaming.

**Results.** As shown in Tab. 2, TideGS incurs modest overhead in the in-memory regime: with Shuffle, it achieves 3.24 img/s, within 1.2% of Naive Offload (3.28 img/s). With Trajectory, TideGS improves to 3.50 img/s and outperforms CLM by 6.4%, indicating that spatiotemporal coherence can offset a substantial part of the virtualization overhead.

*Table 2.* **Throughput in the in-memory regime on Mip-NeRF 360 (avg. over evaluated scenes).** We isolate virtualization overhead in a setting where Native 3DGS fits in VRAM and TideGS blocks are served from memory after warm-up. Native 3DGS (in-memory) serves as the GPU-resident upper bound. TideGS (Shuffle) incurs a small regression due to block management overhead when temporal locality cannot be exploited, while TideGS (Trajectory) leverages spatiotemporal locality to reduce exposed transfer stalls and improves throughput.

| Method | Strategy | Iter (ms)↓ | Img/s↑ | GPU Util. (%)↑ |
|---|---|---|---|---|
| **Native 3DGS** | In-Memory | **253.5** | **3.95** | **96.3** |
| Naive Offload | Shuffle | 305.3 | 3.28 | 74.9 |
| CLM (Zhao et al., 2026) | Shuffle | 304.3 | 3.29 | 66.0 |
| TideGS (Ours) | Shuffle | 309.1 | 3.24 | 62.5 |
| **TideGS (Ours)** | Trajectory | 285.8 | 3.50 | 76.5 |

*Table 3.* **Quality consistency on Mip-NeRF 360.** TideGS preserves Native 3DGS reconstruction quality when scenes are small enough to fit in memory.

| Method | PSNR↑ | SSIM↑ | LPIPS↓ |
|---|---|---|---|
| Native 3DGS (Kerbl et al., 2023) | 29.0252 | 0.8694 | 0.1394 |
| **TideGS (Ours)** | 28.9157 | 0.8689 | 0.1399 |

**Quality consistency with Native 3DGS.** We further verify that parameter virtualization does not materially change reconstruction quality when the scene fits in memory. On the seven public Mip-NeRF 360 scenes, TideGS reaches quality comparable to Native 3DGS (Tab. 3), with a PSNR gap of only 0.11 dB and nearly identical SSIM/LPIPS. This confirms that the main effect of TideGS in the in-memory regime is systems overhead rather than a change to the underlying 3DGS objective.

### 4.2.3. EFFICIENCY IN THE OUT-OF-CORE REGIME

We evaluate out-of-core training on MatrixCity at two primitive scales: the original $\sim$102M Gaussians and a synthetic $\sim$1.1B-Gaussian upscaled version ($10\times$ density). Tab. 4 summarizes throughput and cross-tier traffic.

**Standard scale ($\sim$102M).** Naive Offload runs out of memory, indicating that $\sim$102M Gaussians exceed the practical VRAM budget under per-iteration parameter residency. CLM offloads attributes to system memory but incurs substantial PCIe traffic (0.41 GB/iter), resulting in 100.8 ms/iter. In contrast, TideGS reduces PCIe traffic by $4\times$ to 0.10 GB/iter through trajectory-aware ordering and block reuse, reaching 90.7 ms/iter.

**Billion scale ($\sim$1.1B).** At $\sim$1.1B Gaussians, CLM fails with OOM because it requires GPU-resident geometry, which would demand $\sim$45 GB at this scale. TideGS is the only evaluated single-GPU method that remains feasible and successfully trains the 1.1B scene. As the visible block working set grows with the $10\times$ density increase, PCIe traffic rises to 0.97 GB/iter. Even at this scale, TideGS maintains 49.5% average GPU utilization with 525.6 ms/iter, suggesting that the pipeline does not collapse into an I/O-only

*Table 4.* **Scalability and efficiency on MatrixCity.** We evaluate methods on MatrixCity (∼102M) and its 10× density upscaled version (∼1.1B). CLM runs at ∼102M but fails with **OOM** at ∼1.1B due to its GPU-resident geometry requirement (∼45 GB). TideGS scales to ∼1.1B on a single 24 GB GPU by keeping PCIe traffic proportional to the streamed working-set delta.

| Method | Scale ($N$) | Backing Store | PCIe (GB/iter)↓ | GPU Util. (%)↑ | Iter (ms)↓ |
|---|---|---|---|---|---|
| Naive Offload | ∼102M | System RAM | *— Out of Memory (OOM) —* | | |
| CLM (Zhao et al., 2026) | ∼102M | System RAM | 0.41 | 37.0 | 100.8 |
| **TideGS (Ours)** | ∼102M | NVMe SSD | **0.10** | 43.3 | **90.7** |
| CLM (Zhao et al., 2026) | ∼1.1B | System RAM | *— Out of Memory (OOM) —* | | |
| **TideGS (Ours)** | ∼1.1B | NVMe SSD | 0.97 | **49.5** | 525.6 |

execution mode. This is consistent with TideGS overlapping SSD reads/prefetch, H2D transfers, and GPU computation (Sec. 3.4), which hides a substantial fraction of SSD/PCIe latency.

### 4.2.4. QUALITY SCALING

We study how reconstruction quality scales with the number of Gaussians ($N$) on the test split of the **MatrixCity BigCity/Aerial** subset. To decouple capacity from adaptive model growth, we disable densification and pruning in all settings and evaluate fixed-size initializations at three scales: **Standard** (∼25M), **Large** (∼102M), and **Billion** (∼1.1B).

**Fixed-size initialization from RGB-D backprojection.** All scales share the same initialization pipeline and differ only in the number of retained primitives. We backproject RGB-D observations from the MatrixCity BigCity/Aerial training split (51,623 images) into a colored point cloud using the provided camera intrinsics/extrinsics. We generate a ∼1B-point initialization by (i) downsampling images by a factor `ds` during projection and (ii) uniformly subsampling valid depth points with stride `ratio`. We then derive the 102M and 25M initializations by uniform downsampling of the 1B point cloud, ensuring that all three scales are sampled from the same underlying geometry distribution.

**Training protocol.** Across all scales, we keep the training recipe identical (train/test splits, optimizer and learning-rate schedule, batch size, and rendering resolution). Therefore, differences in Fig. 4 primarily reflect the effect of model capacity rather than changes in training configuration.

**Results.** As shown in Fig. 4, Native 3DGS cannot reach the Standard scale under our 24 GB setting due to full-state VRAM residency. CLM trains at ∼102M and reaches 25.0 dB PSNR, but fails with OOM at ∼1.1B Gaussians. In contrast, **TideGS** is the only evaluated single-GPU method that trains at the billion scale, and the added capacity translates into higher fidelity: at ∼1.1B Gaussians, TideGS achieves **26.1 dB** PSNR, improving over the largest feasible baseline.

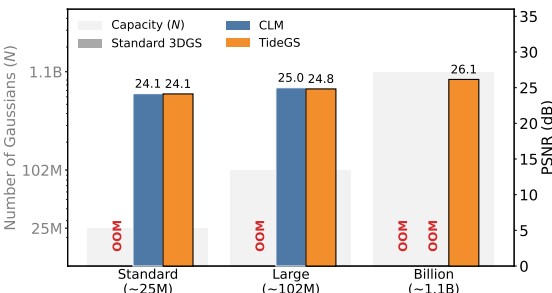

*Figure 4.* **Quality scaling on MatrixCity.** PSNR across different Gaussian counts ($N$). "Standard", "Large", and "Billion" correspond to ∼25M, ∼102M, and ∼1.1B Gaussians, respectively. Native 3DGS and CLM encounter OOM at higher scales, while **TideGS** enables billion-scale training on a single GPU and achieves the highest PSNR among evaluated methods (26.1 dB).

### 4.3. Ablation Study

To validate the contribution of each system component, we conduct an ablation study on the MatrixCity dataset (∼102M). Measurements are reported at a rendering resolution of 1920×1080 and averaged over 500 iterations with a prefetch batch size of 64 frames. We choose the 102M scale because it is large enough to stress the I/O subsystem while still allowing the unoptimized variants to run, making numerical comparisons meaningful. Tab. 5 summarizes the impact of removing key designs.

**Impact of Differential Streaming (w/o Tide).** Disabling the differential streaming policy forces the system to retransmit all visible blocks per iteration, regardless of their residency status in VRAM. This increases PCIe traffic by **8.5×**, from 0.10 GB/iter to 0.85 GB/iter. The larger transfer volume makes training more bandwidth-bound and increases iteration latency to 145.3 ms, confirming that residency-aware differential streaming is important for minimizing cross-tier communication.

**Impact of Asynchronous Pipeline (w/o Overlap).** In this ablation, we keep the *same* working set and transfer volume per iteration (hence similar PCIe traffic), but disable overlap between data movement and GPU compute by serializing SSD read/prefetch, H2D staging, and rendering/backpropagation. Without concurrent prefetching/staging for the next iteration, SSD/PCIe latency is exposed on the critical path, forcing the GPU to wait and increasing iteration time to **210.5 ms**. Compared with the full method (90.7 ms), this result shows that overlapping transfers with compute is essential to reduce exposed I/O stalls.

**Impact of Spatial Locality (w/o Morton).** Replacing our Morton-ordered block layout with a random arrangement removes geometric coherence and increases working-set churn. This causes the CPU cache hit rate to drop from 95.2% to **42.1%**, leading to frequent evictions and

*Table 5.* **Component Ablations on MatrixCity (∼102M).** We evaluate the impact of key system designs. **w/o Tide:** Disabling differential streaming sharply increases PCIe traffic. **w/o Overlap:** Serializing data movement and compute exposes I/O latency, increasing iteration time. **w/o Morton:** Random ordering removes locality and substantially reduces cache hit rates.

| Variant | Iter (ms)↓ | PCIe (GB/iter)↓ | CPU Cache Hit (%)↑ |
|---|---|---|---|
| **Full TideGS** | **90.7** | **0.10** | **95.2** |
| w/o Tide (Diff. Stream) | 145.3 | 0.85 | 95.2 |
| w/o Overlap (Async) | 210.5 | 0.10 | 95.2 |
| w/o Morton (Locality) | 115.8 | 0.45 | 42.1 |

*Table 6.* **Quality metrics for core system ablations.** The main system components affect data-movement efficiency while preserving reconstruction quality.

| Variant | PSNR↑ | SSIM↑ | LPIPS↓ | Iter (ms)↓ | PCIe (GB/iter)↓ |
|---|---|---|---|---|---|
| **Full TideGS** | 24.83 | 0.71 | 0.39 | **90.7** | **0.10** |
| w/o Overlap (Async) | 24.86 | 0.71 | 0.39 | 210.5 | 0.10 |
| w/o Tide (Diff. Stream) | 24.79 | 0.71 | 0.39 | 145.3 | 0.85 |

re-fetches. Consequently, the system transfers substantially more data per iteration (PCIe traffic rises to 0.45 GB/iter), degrading throughput. This validates that preserving spatial locality is a prerequisite for efficient out-of-core traversal.

**Quality impact.** We also evaluate final reconstruction metrics for the communication and pipelining ablations. As shown in Tab. 6, disabling differential streaming or overlap primarily affects efficiency rather than final quality: PSNR, SSIM, and LPIPS remain nearly unchanged across these variants. This is expected because these components preserve the same visible Gaussian set and optimization objective while changing how data movement is scheduled.

## 5. Related Work

Prior work improves 3DGS scalability from several complementary directions. *Distributed and scene-partitioned scaling* increases capacity by either spreading parameters across multiple GPUs (Zhao et al., 2025; Li et al., 2024; Tao et al., 2025; Haberl et al., 2025; Gao et al., 2025) or decomposing a large scene into independently trained regions (Liu et al., 2025b;c;a; Chen & Lee, 2024; Lin et al., 2024). These approaches can scale scene size, but they rely on additional GPU memory, interconnects, or explicit boundary handling. TideGS targets a different operating point: it keeps training on a single GPU and expands effective capacity through an SSD–CPU–GPU memory hierarchy.

*Compression, pruning, and kernel optimizations* reduce the memory or compute cost of 3DGS by pruning/compressing Gaussians (Hanson et al., 2025; Tian et al., 2025; Fang & Wang, 2024; Mallick et al., 2024; Lu et al., 2024) or improving rasterization and scheduling efficiency (Liao et al., 2025; Gui et al., 2024; Durvasula et al., 2025; Feng et al., 2025; Höllein et al., 2025). These techniques are valuable and

largely orthogonal to TideGS, but many primarily benefit inference or per-iteration throughput rather than eliminating the training-time residency requirement for parameters, gradients, and optimizer states. In contrast, TideGS virtualizes the training state itself and materializes only the visible working set needed by each iteration.

*Host-offloading and hierarchical training systems* are closest to our work. GS-Scale (Lee et al., 2026) and CLM (Zhao et al., 2026) move parameters or optimizer states to CPU memory, but still retain key geometry or rasterization-dependent state in VRAM, leaving scalability bounded by GPU-resident data at large $N$. More general systems, such as ZeRO-style partitioning (Rasley et al., 2020; Ren et al., 2021) and embedding-table caching (Wilkening et al., 2021; Song et al., 2023), offer hierarchy-management ideas but do not exploit 3DGS visibility sparsity or camera-trajectory locality. TideGS extends offloading to SSD-backed parameter storage with block-virtualized geometry, asynchronous SSD–CPU–GPU execution, and trajectory-adaptive differential streaming, so VRAM caches the sparse active working set rather than persisting the full parameter store.

## 6. Conclusion

We presented TideGS, an out-of-core training framework for 3D Gaussian Splatting that virtualizes the full Gaussian parameter table across an SSD–CPU–GPU hierarchy and materializes only the per-iteration working set on GPU. By combining block-virtualized geometry, asynchronous cross-tier execution, and trajectory-adaptive differential streaming, TideGS shifts the single-GPU bottleneck from persistent VRAM residency to locality-aware working-set management. Experiments show that TideGS trains a 1.1B-Gaussian MatrixCity scene on a single 24 GB GPU, preserves Native 3DGS quality in the in-memory regime, and improves city-scale reconstruction fidelity, suggesting that out-of-core optimization can make large-scale 3DGS training more accessible.

**Limitations.** TideGS relies on camera-order locality; unstructured image collections can reduce differential-streaming reuse and increase block churn and cross-tier traffic. Throughput also depends on the SSD tier: slower storage than our 3.3 GB/s NVMe operating point and append-only logging can raise latency, temporary footprint, and endurance pressure, though compaction can run off the critical path. TideGS discards Adam moments for evicted blocks, so high churn can cause optimizer-state cold starts; larger CPU caches or selective moment persistence may help. It complements distributed in-memory multi-GPU training: the latter can improve throughput with sufficient hardware, while TideGS lowers the hardware floor for billion-scale training on a single GPU.

## Acknowledgements

The authors thank Sixu Li for helpful suggestions and assistance with manuscript writing and proofreading. This research is supported by HKUST Start-up grants R9895 and R9109 from CSE, and RGC-NSFC project CRS HKUST601/24.

## Impact Statement

This paper presents work whose goal is to advance the field of Machine Learning. There are many potential societal consequences of our work, none of which we feel must be specifically highlighted here.

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

# A. Appendix

## A.1. Additional Experimental Details

**Hardware and software.**   Unless otherwise noted, experiments are conducted on a single workstation with one NVIDIA RTX A5000 GPU (24 GB VRAM), an AMD EPYC 7532 CPU (32 cores), and 256 GB DDR4 system memory. Out-of-core storage uses a Samsung PM9A3 enterprise NVMe SSD (PCIe Gen4 ×4; measured I/O speed 3.3 GB/s) formatted with ext4. TideGS is implemented in PyTorch 2.6.0 with CUDA 12.4.

**Controlling OS caching effects.**   OS page caching can affect repeated SSD-backed measurements by serving recently accessed pages from DRAM. To reduce this confound, compared runs use fresh run/cache directories and avoid reusing warm SSD patch or application-cache state. For standalone cold-cache SSD bandwidth measurements, we use a cold-start protocol to evict cached pages before measurement.

**Baselines.**   Native 3DGS (Kerbl et al., 2023) is the official implementation that keeps model parameters, gradients, and optimizer states in GPU memory. Naive Offload is a ZeRO-Offload-inspired (Ren et al., 2021) host-offloading baseline that keeps optimizer states and gradients in CPU memory, materializes the full Gaussian parameter table on GPU for each iteration, stores gradients back to CPU, and performs Adam updates on CPU. CLM (Zhao et al., 2026) is a state-of-the-art host-offloading pipeline that offloads high-dimensional attributes to CPU memory while keeping key geometry required by the rasterizer resident in GPU memory. We do not compare to multi-GPU systems (e.g., RetinaGS (Li et al., 2024), Grendel-GS (Zhao et al., 2025)) in the main experiments because they rely on aggregate device memory and interconnects beyond our single-GPU setting.

**Preprocessing and metrics.**   The block layout is built once before training by Morton sorting, computing block bounds/statistics, and writing the initial base segment to SSD. This preprocessing takes 1.9 minutes at ∼102M Gaussians and 21.2 minutes at ∼1.1B Gaussians on MatrixCity, accounting for less than 0.5% of total training time in both settings. We report four classes of metrics: **quality** (PSNR, SSIM, and LPIPS on held-out views when applicable), **throughput** (average iteration time and/or images/s measured over a fixed window after warm-up), **system** (PCIe traffic in GB/iter, SSD read/write throughput in GB/s, prefetch/cache hit rate, and GPU utilization as the time-averaged `utilization.gpu` from `nvidia-smi` sampled at 1 Hz), and **resources** (VRAM/DRAM usage and SSD footprint).

## A.2. Ordering and Convergence Discussion

TideGS departs from the standard randomized view order used in 3DGS by processing views in a trajectory-aware order. We do *not* aim to prove a new convergence theorem for masked adaptive optimizers. Instead, we provide an intuition consistent with ordering-aware SGD analyses (Mohtashami et al., 2022) and with the two empirical properties exploited by TideGS: (i) **visibility-induced sparsity**, where each iteration updates only a small subset of Gaussians, and (ii) **trajectory continuity**, where consecutive views along a smooth camera path tend to have similar visibility and gradients.

### A.2.1. PROBLEM SETUP AND NOTATION

Let $\theta \in \mathbb{R}^d$ denote the concatenation of all Gaussian parameters across all primitives. Given $M$ training views, the objective is

$$\min_{\theta} \ F(\theta) := \frac{1}{M} \sum_{i=1}^{M} f_i(\theta), \tag{6}$$

where $f_i(\theta)$ is the photometric (rendering) loss for view $i$.

**Spatial blocks for storage and streaming.**   Gaussians are partitioned into *spatial blocks* using Morton-code ordering (Sec. 3.3): each block contains a fixed number of primitives (e.g., $B$ Gaussians) and serves as the basic unit for SSD–CPU–GPU streaming. Let $K$ be the number of blocks and let the block-index universe be $\mathcal{K} = \{0, \ldots, K-1\}$. We use $\mathcal{K}_t \subseteq \mathcal{K}$ for the coarse block working set selected by block-wise frustum culling at iteration $t$.

**Trajectory ordering over views.**   Unlike standard 3DGS, which samples views with random shuffling, TideGS constructs an ordered sequence of training views by applying a clustered traveling-salesperson (TSP) ordering over camera poses. Let $\pi = (\pi_1, \ldots, \pi_M)$ be the resulting permutation of *views* (not spatial blocks), and the training loop processes views in this

order. This preserves the same empirical objective because each view is still visited once per pass through the training set, but changes the order in which views are presented to the optimizer.

**Visibility-induced sparse updates.** At iteration $t$, only Gaussians that are visible after coarse-to-fine filtering receive gradients. Let $\mathcal{I}_t$ denote the Gaussian-level active set, i.e., the Gaussian indices and parameter coordinates updated at iteration $t$. In TideGS, the optimizer update is *masked*:

$$\theta_{t+1}^{(j)} = \begin{cases} \theta_t^{(j)} - \eta_t \cdot u_t^{(j)}, & j \in \mathcal{I}_t, \\ \theta_t^{(j)}, & \text{otherwise,} \end{cases} \tag{7}$$

where $u_t^{(j)}$ denotes the optimizer's update direction, e.g., Adam using first and second moments. This captures the key fact used by TideGS: most parameters are untouched at each step.

### A.2.2. EMPIRICAL INTUITIONS

We use two TideGS-specific empirical properties to explain why trajectory ordering is stable in practice.

**Property 1 (Localized updates under visibility sparsity).** Views that are close in space tend to activate nearby spatial blocks, while parameters far from the current camera are inactive. Thus, active sets are localized: distant regions receive little interference from unrelated views, and adjacent views tend to induce overlapping active sets. We use this property only as an intuition for why masked 3DGS updates are less exposed to harmful cross-region interference.

**Property 2 (Trajectory continuity reduces gradient variation).** For a trajectory-ordered permutation $\pi_{\text{TSP}}$, consecutive views are geometrically close, so their per-view gradients are more similar on the relevant active coordinates than under a random order. This behavior can be summarized by a smaller *gradient-variation* surrogate, defined below, for $\pi_{\text{TSP}}$ than for random permutations.

### A.2.3. TWO INTUITIONS: SPARSITY LIMITS STALENESS; ORDERING LIMITS VARIATION

**Visibility sparsity limits the propagation of stale optimizer state.** The primary concern with non-shuffled training is that sequentially correlated samples may introduce bias or cause optimizer state, e.g., momentum, to become stale or misaligned. In TideGS, the masked update in Eq. (7) provides a natural safeguard: optimizer state is updated only on active coordinates $\mathcal{I}_t$. Thus, parameters that are not visible for long stretches are not repeatedly perturbed by unrelated views, and their update history is dominated by iterations where they are visible. This visibility-induced sparsity can reduce interference across distant spatial regions and make the optimization more locally structured.

**Trajectory ordering reduces the ordering-dependent variation term (informal).** Ordering-aware SGD analyses, including random reshuffling, suggest that the effect of using a fixed permutation can be characterized by an ordering-dependent term related to how rapidly per-sample gradients change along the permutation (Mohtashami et al., 2022). We use the following surrogate to capture this intuition:

$$V_\pi(\{\theta_t\}) := \sum_{t=1}^{M-1} \left\| \nabla f_{\pi_t}(\theta_t) - \nabla f_{\pi_{t+1}}(\theta_t) \right\|^2. \tag{8}$$

We use $V_\pi$ only as a qualitative surrogate of ordering-induced variation, evaluated along the training trajectory $\{\theta_t\}$ in the same spirit as ordering-aware analyses that relate error constants to permutation-dependent gradient drift. In TideGS, this variation is most relevant on active coordinates (roughly $\mathcal{I}_t \cup \mathcal{I}_{t+1}$), since inactive parameters are not updated. For random permutations, consecutive views are typically unrelated, yielding larger variation. For $\pi_{\text{TSP}}$, consecutive views are geometrically adjacent; under the trajectory-continuity intuition in Property 2, this suggests smaller gradient changes between consecutive steps, i.e., $V_{\pi_{\text{TSP}}} \ll V_{\pi_{\text{Random}}}$ in practice. Intuitively, the trajectory order makes the optimization process "locally consistent" across steps, which can help compensate for the lack of global shuffling.

### A.2.4. TAKEAWAY: WHY TRAJECTORY ORDERING IS STABLE IN TIDEGS

Putting the above together: (1) TideGS updates only the Gaussian-level active set $\mathcal{I}_t$ at each iteration, which limits cross-region interference and reduces the impact of stale optimizer state on inactive parameters; (2) trajectory-aware ordering

makes consecutive views similar, reducing the ordering-dependent gradient variation captured by Eq. (8). Therefore, although TideGS departs from standard randomized view ordering, its training sequence remains stable in practice due to the *combination* of visibility sparsity and trajectory continuity, consistent with the empirical convergence and quality results in Sec. 4.

### A.3. Ordering Ablation: Shuffle vs. Trajectory

To quantify the effect of trajectory ordering on optimization quality, we compare TideGS with randomized view shuffling and trajectory-ordered views on the *bicycle* scene from Mip-NeRF 360, which fits in GPU memory. Both variants use the same training views, loss, and optimization recipe; only the view presentation order differs. As shown in Tab. 7, trajectory ordering improves iteration time while maintaining similar final reconstruction quality.

*Table 7.* **Shuffle vs. trajectory ordering on Mip-NeRF 360 *bicycle*.** Trajectory ordering improves locality and reduces iteration time, with a small quality gap relative to randomized shuffling.

| Method | Ordering | Final PSNR↑ | Final SSIM↑ | Final LPIPS↓ | Iter (ms)↓ |
|---|---|---|---|---|---|
| TideGS | Shuffle | 24.8595 | 0.7226 | **0.2932** | 392.36 |
| **TideGS** | Trajectory | 24.6281 | 0.7213 | 0.3063 | **339.86** |

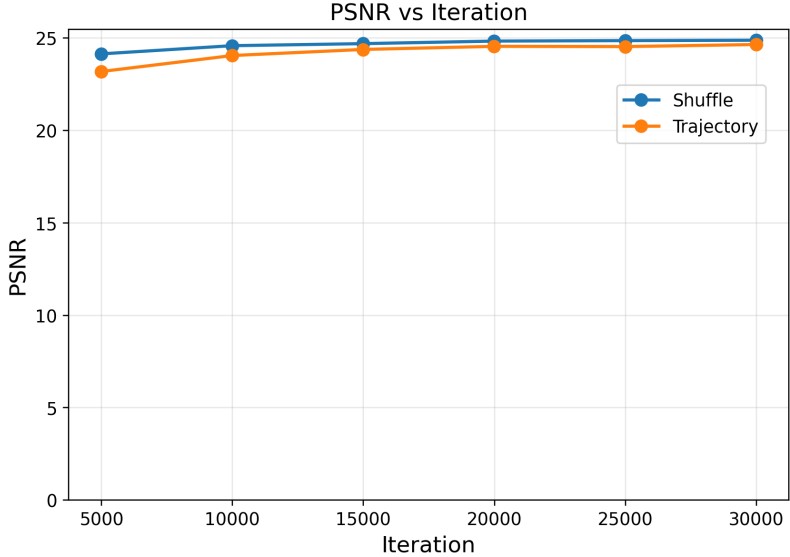

*Figure 5.* **Convergence under different view orders.** The curves compare randomized shuffling and trajectory ordering on Mip-NeRF 360 *bicycle*, complementing the final metrics in Tab. 7.

### A.4. Dense Initialization Without Training-Time Densification

Our large-scale MatrixCity experiments use fixed-size initializations and disable densification and pruning to isolate out-of-core memory management from adaptive model growth. To check whether this design choice materially changes reconstruction quality, we compare dense-initialized fixed-size training with the standard densification setting on two representative Mip-NeRF 360 scenes. As shown in Figs. 6 and 7, the fixed-size dense initialization reaches similar final quality to the densify-on setting when initialized with a comparable number of primitives (32.29 vs. 32.02 PSNR on *bonsai*, and 25.26 vs. 25.23 PSNR on *bicycle*).

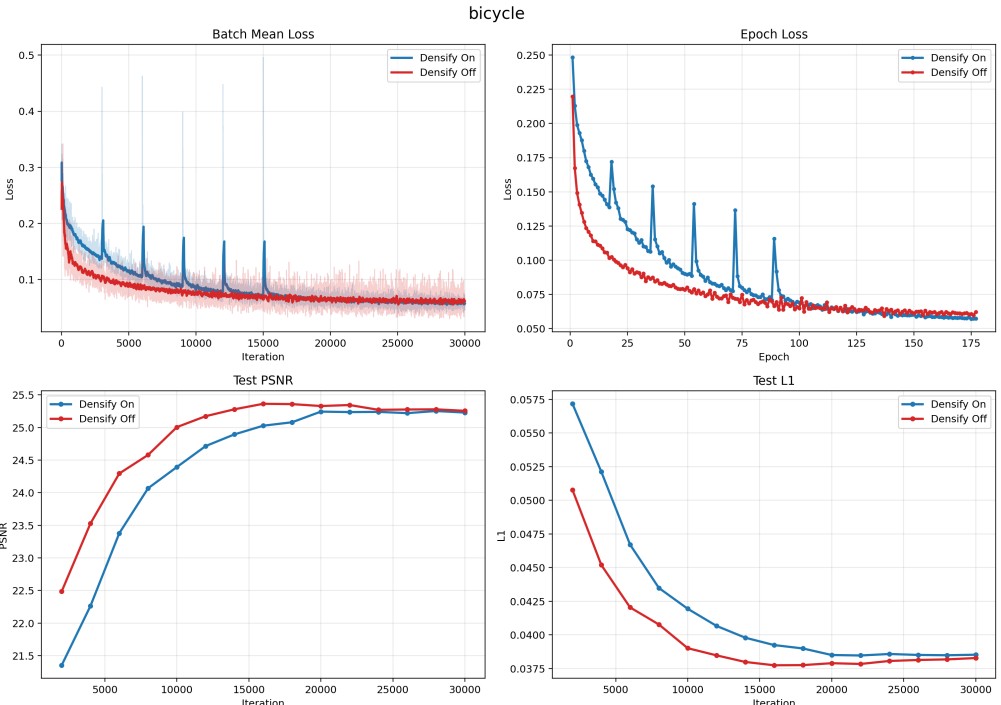

*Figure 6.* **Dense initialization versus training-time densification on *bicycle*.** Dense-initialized fixed-size training achieves comparable final quality to the standard densification setting on this outdoor scene.

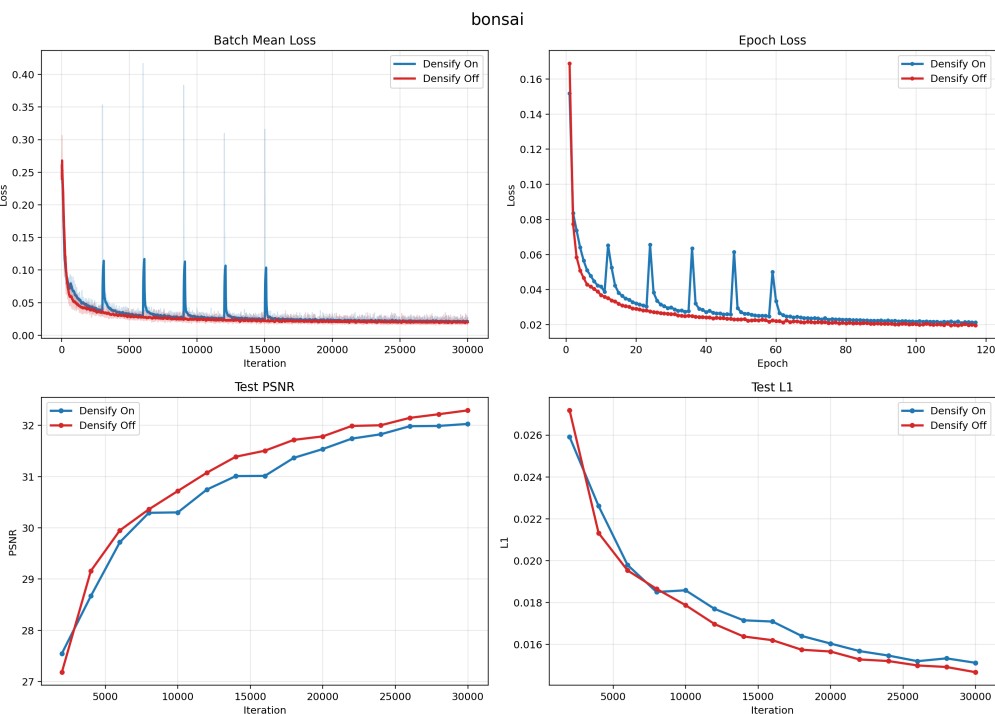

*Figure 7.* **Dense initialization versus training-time densification on *bonsai*.** Dense-initialized fixed-size training achieves comparable final quality to the standard densification setting on this indoor scene.

## A.5. Additional System Measurements

We collect additional system measurements that complement the main experiments: hardware validation beyond the A5000 setting and optimizer-state churn under out-of-core residency.

**Additional hardware validation.** Tab. 8 extends the MatrixCity scalability measurements in Tab. 4 to an RTX 3090 and an A800. The results show the same qualitative behavior as on the A5000: TideGS reduces PCIe traffic and iteration time at the ∼102M-Gaussian scale, and remains feasible at the ∼1.1B-Gaussian scale where CLM runs out of memory.

*Table 8.* **Additional hardware validation on MatrixCity.** TideGS exhibits the same scaling trend on an RTX 3090 and an A800, showing that the observed benefits are not specific to the A5000 used in the main experiments.

| GPU | Method | Scale | PCIe (GB/iter)↓ | GPU Util. (%)↑ | Iter (ms)↓ |
|---|---|---|---|---|---|
| RTX 3090 | Naive Offload | ∼102M | — Out of Memory (OOM) — | | |
| RTX 3090 | CLM (Zhao et al., 2026) | ∼102M | 0.43 | 37.6 | 95.2 |
| RTX 3090 | **TideGS (Ours)** | ∼102M | **0.12** | **43.6** | **89.3** |
| A800 | Naive Offload | ∼102M | 0.76 | 50.4 | 156.1 |
| A800 | CLM (Zhao et al., 2026) | ∼102M | 0.38 | 42.2 | 91.8 |
| A800 | **TideGS (Ours)** | ∼102M | **0.13** | **50.3** | **77.3** |
| A800 | CLM (Zhao et al., 2026) | ∼1.1B | — Out of Memory (OOM) — | | |
| A800 | **TideGS (Ours)** | ∼1.1B | 1.0 | **55.8** | 498.9 |

**Optimizer-state churn.** Tab. 9 reports the residency and cold-restart statistics behind the optimizer-state placement design in Sec. 3.5. These measurements quantify how often blocks are evicted, re-admitted, and re-initialized under the evaluated settings.

*Table 9.* **Residency and cold-restart statistics.** We report block churn and optimizer-state cold-start rates under the evaluated settings.

| Scale | Block eviction rate | Re-admission rate | Cold-restarted updates / total updates | Mean resident streak |
|---|---|---|---|---|
| ∼102M Gaussian primitives | 4.8% | 1.4% | 0.6% | 18.7 iters |
| ∼1.1B Gaussian primitives | 6.9% | 2.1% | 0.9% | 13.4 iters |

## A.6. Single-GPU vs. Distributed In-Memory Operating Points

TideGS targets a different operating point from distributed in-memory systems such as Grendel-GS (Zhao et al., 2025) and RetinaGS (Li et al., 2024). Distributed training can reduce wall-clock time when multiple GPUs, aggregate VRAM, and interconnect bandwidth are available, while TideGS lowers the hardware floor by trading additional storage hierarchy management for single-GPU scalability. Tab. 10 provides a bill-of-materials-style capacity-per-dollar comparison to contextualize this trade-off. Figs. 8 and 9 further compare TideGS with Grendel-GS (Zhao et al., 2025) from wall-clock and iteration-wise convergence perspectives.

*Table 10.* **Capacity-per-dollar operating point.** The comparison contextualizes TideGS against a representative 4-GPU distributed in-memory setup; it is intended as an operating-point comparison rather than a claim of universal superiority.

| System | Node cost | Max trainable Gaussian primitives | Gaussian primitives / USD |
|---|---|---|---|
| TideGS node | $6,525.95 | 1.0B | 153,000 |
| 4-GPU Grendel-GS-style node | $15,055.98 | 44.9M | 2,980 |

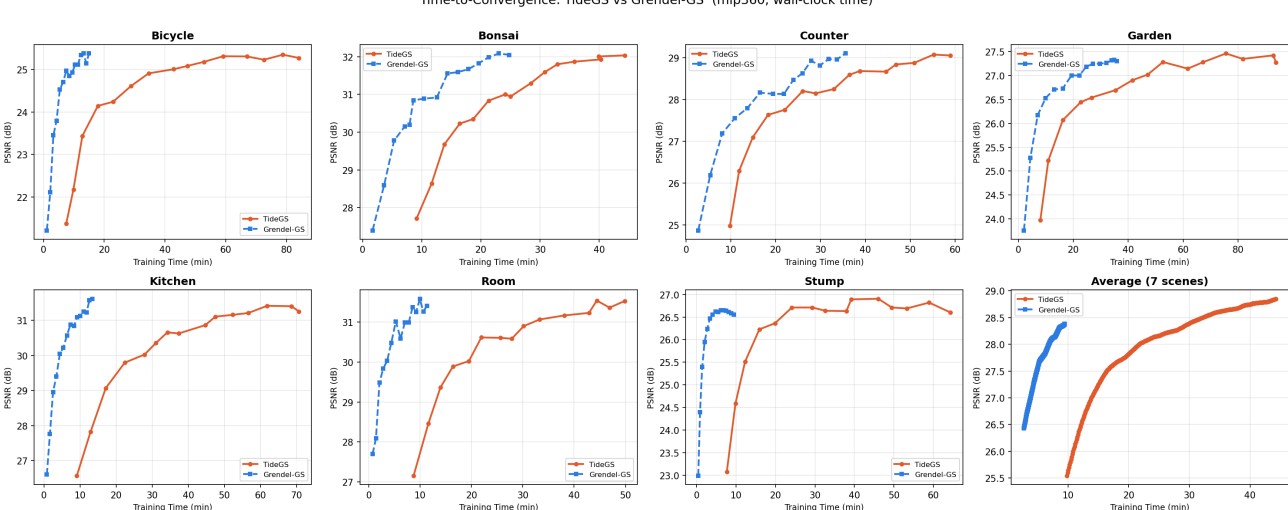

*Figure 8.* **Wall-clock time-to-convergence operating point.** We compare TideGS with the distributed in-memory baseline, Grendel-GS (Zhao et al., 2025), from the wall-clock perspective to contextualize the trade-off between hardware scale and single-GPU out-of-core capacity.

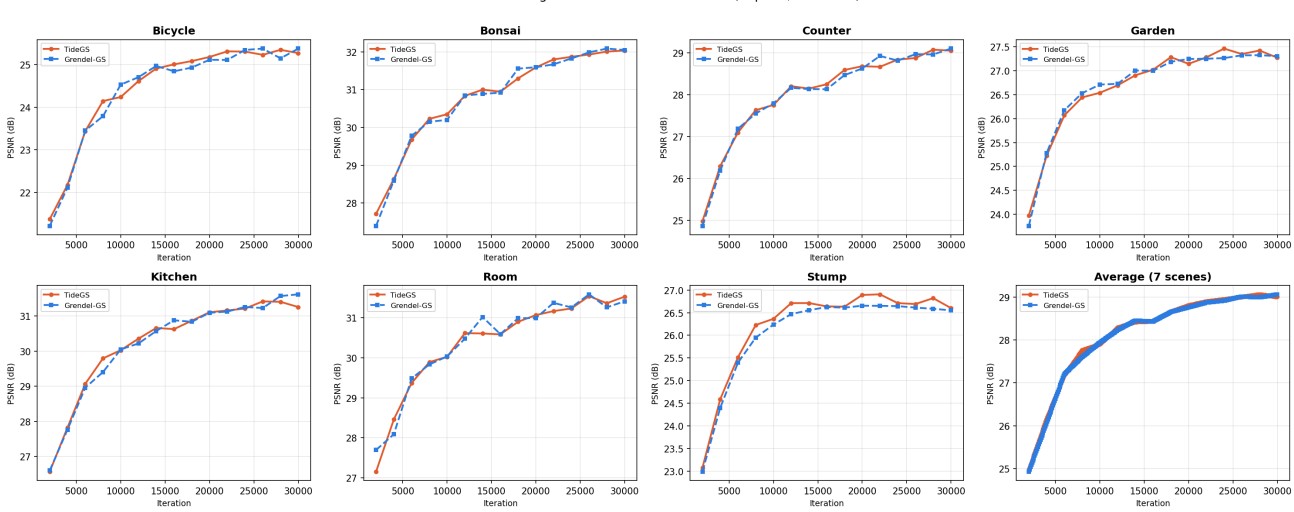

*Figure 9.* **Iteration-wise convergence operating point.** We compare TideGS with the distributed in-memory baseline, Grendel-GS (Zhao et al., 2025), from the training-iteration perspective to separate optimization progress from per-iteration execution time.

