# OpenReview forum: "TideGS: Scalable Training of Over One Billion 3D Gaussian Splatting Primitives via Out-of-Core Optimization"
_ICML.cc/2026/Conference — ICML 2026 spotlight_

### Official Review · Reviewer_po1s · 2026-03-12

**Soundness:** 3
**Presentation:** 3
**Significance:** 3
**Originality:** 3
**Overall Recommendation:** 4
**Confidence:** 4

**Summary:**

This paper addresses the scalability limitations of 3D Gaussian Splatting (3DGS) training, which is typically constrained by GPU memory because each Gaussian primitive contains a relatively large set of parameters. As scene complexity increases, the number of Gaussians required for high-quality reconstruction quickly exceeds the capacity of commodity GPUs, limiting existing systems to tens of millions of primitives.

The authors propose **TideGS**, an out-of-core training framework that enables training of extremely large 3DGS models (up to billion-scale Gaussians) on a single consumer GPU. The key insight is that 3DGS optimization is inherently sparse: at each iteration only the Gaussians visible from the current camera batch are actively used. Therefore, GPU memory can be treated as a working-set cache instead of storing the entire parameter table.

Based on this observation, the paper introduces three main system components:

1. **Block-virtualized geometry**, which organizes Gaussian primitives into spatial blocks aligned with SSD storage to improve data locality and streaming efficiency.
2. **A hierarchical asynchronous pipeline** spanning SSD, CPU memory, and GPU memory to overlap data movement with rendering and optimization.
3. **Trajectory-adaptive differential streaming**, which transfers only incremental updates to the active working set across iterations to reduce data transfer overhead.

Experiments demonstrate that TideGS can scale training to over one billion Gaussians using a single consumer GPU while achieving high-quality scene reconstruction. The method outperforms previous out-of-core baselines and conventional in-memory 3DGS training systems in both scalability and reconstruction fidelity on large-scale scenes.

**Compliance With Llm Reviewing Policy:**

Affirmed.

**Final Justification:**

I would recommend accept this paper, my concerns have been addressed in the rebuttal.

**Key Questions For Authors:**

1. **Provides Visual Quality Metrics in Ablation Study:**
   Can authors provide visual metrics such as PSNR, SSIM, LPIPS in ablation result?

2. **Storage bandwidth sensitivity:**
   How sensitive is the method to SSD bandwidth and latency? Would performance degrade significantly on lower-end storage devices?

3. **Training throughput comparison:**
   How does the training time per iteration compare to standard in-memory 3DGS training when the scene size is within GPU memory limits?

4. **Memory management strategy:**
   How is the working set of active Gaussians predicted or scheduled across iterations? Is it based purely on camera trajectory, or are there additional heuristics?

**Limitations:**

Limitations related to storage bandwidth dependence and the system-specific nature of the framework. The method's performance may depend on hardware characteristics such as SSD bandwidth.

**Strengths And Weaknesses:**

Strengths

1. **Addresses an important systems bottleneck.**
   The paper tackles a fundamental limitation of current 3D Gaussian Splatting systems: the inability to scale beyond GPU memory capacity. This problem is practically important as scenes become increasingly large and detailed, especially for applications such as city-scale reconstruction and large environment modeling.

2. **Clear systems insight.**
   The key observation that only a subset of Gaussians is active at each training iteration is well motivated and provides a principled foundation for designing an out-of-core training pipeline. Treating GPU memory as a working-set cache is a compelling idea that aligns with memory hierarchy principles.

3. **Well-structured system design.**
   The proposed framework combines several complementary components—block-virtualized storage, asynchronous hierarchical pipelines, and differential streaming—to reduce I/O overhead and maintain training throughput. The integration of these mechanisms appears technically sound and demonstrates thoughtful system engineering.

4. **Significant scalability improvements.**
   The reported ability to train with over one billion Gaussians on a single consumer GPU represents a substantial improvement compared to previous systems that operate at the scale of tens of millions of primitives. This suggests the method may enable new applications involving very large scenes.

Weaknesses

1. **Limited methodological novelty.**
   While the system design is well executed, many of the individual components (hierarchical memory pipelines, asynchronous I/O, and spatial partitioning) resemble established techniques from database and systems research. The primary contribution lies in their adaptation to the 3DGS training pipeline rather than introducing fundamentally new algorithms.

2. **Potential I/O bottleneck concerns.**
   The approach relies heavily on SSD streaming and hierarchical data movement. It would be useful to understand how performance varies under different storage bandwidth conditions and whether the method remains practical on slower storage devices.

3. **Clarity of some system details.**
   Certain implementation details, such as the scheduling strategy for asynchronous streaming and the exact block management mechanism, could be explained more clearly to facilitate reproducibility.

---

> ### Author Rebuttal · Authors · 2026-03-31
>
> We thank the reviewer for the positive evaluation and helpful feedback.
> We appreciate these comments, and the added discussion, tables, and analysis described below will be incorporated into the final version to further strengthen the paper.
>
> **W1. Limited methodological novelty**
>
> We humbly clarify that our intended contribution is not a new low-level I/O primitive in isolation, nor a new neural rendering algorithm, but a 3DGS-specific reformulation and system co-design that makes out-of-core training practical. The key observation is that 3DGS training is visibility-sparse and trajectory-conditioned, so GPU memory can be treated as a working-set cache while the full parameter table is virtualized across the SSD-CPU-GPU hierarchy rather than kept persistently in device memory.
>
> Built on this view, TideGS combines block-virtualized geometry, trajectory-adaptive differential streaming, and an asynchronous SSD-CPU-GPU pipeline in a way tailored to 3DGS rendering/backpropagation. We believe this is why the design has been viewed by other reviewers as "quite simple, but makes sense and feasible" and as "a highly effective system": the contribution is a simple but non-trivial systems synthesis for 3DGS.
>
> More broadly, simple and effective systems designs are also appreciated at ICML; for example, FlexGen (ICML 2023) shows that a well-motivated systems co-design is itself a meaningful contribution.
>
> **Q1. Provides visual quality metrics in ablation study**
>
> Following the reviewer’s suggestion, we extend Tab. 4 of the main paper, "Component Ablations on MatrixCity," by adding PSNR / SSIM / LPIPS:
>
> | Variant | PSNR | SSIM | LPIPS | Iter time | PCIe traffic |
> | --- | --- | --- | --- | --- | --- |
> | Full TideGS | 24.83 | 0.71 | 0.39 | 90.7 ms | 0.10 GB/it |
> | w/o overlap | 24.86 | 0.71 | 0.39 | 210.5 ms | 0.10 GB/it |
> | w/o differential streaming | 24.79 | 0.71 | 0.39 | 145.3 ms | 0.85 GB/it |
>
> These results show that the ablated variants mainly affect efficiency rather than final reconstruction quality, which is consistent with their role as systems optimizations.
>
> **W2 / Q2. Storage bandwidth sensitivity**
>
> The performance of TideGS does depend on SSD bandwidth and latency, but for most internal NVMe SSDs this is not the dominant bottleneck in practice.
>
> Following the reviewer’s suggestion, we make this storage dependence more explicit. All SSD-backed runs use a real Samsung PM9A3 NVMe SSD with measured read bandwidth of 3.3 GB/s under a cold-start protocol. The current ablation already shows that TideGS depends on reducing and overlapping cross-tier traffic rather than naively streaming from storage: removing overlap increases iteration time from 90.7 ms to 210.5 ms, while removing differential streaming increases traffic from 0.10 GB/iter to 0.85 GB/iter.
>
> Based on the measured metrics in Tab. 3 of the main paper, a simple hide-latency threshold estimate gives about 1.10 GB/s at the ~102M-Gaussian-primitive scale and about 1.85 GB/s at the ~1.1B-Gaussian-primitive scale. In practice, these thresholds are compatible with most internal NVMe SSDs, though generally not with SATA SSDs.
>
> **Q3. Training throughput comparison**
>
> This comparison is already the purpose of Tab. 2 in the main paper, where we quantify TideGS overhead when the scene size is small enough to fit in GPU memory. Native 3DGS runs at 253.5 ms/iter (3.95 img/s), while TideGS runs at 309.1 ms/iter (3.24 img/s) with Shuffle and 285.8 ms/iter (3.50 img/s) with Trajectory. Thus, TideGS incurs only modest in-memory overhead relative to native GPU-resident training, and trajectory-aware scheduling recovers a substantial part of this overhead by exploiting spatiotemporal locality.
>
> **W3 / Q4. Memory management strategy**
>
> In TideGS, the working set is not predicted purely from camera trajectory. Instead, it is determined from the current camera batch through a two-stage visibility process: CPU-side coarse block culling uses per-block bounds to obtain the active block set, and GPU-side fine filtering/rendering determines the final contributing Gaussians. Camera trajectory is then used only to improve cross-iteration reuse: under smooth motion, consecutive active block sets overlap substantially, so TideGS retains the overlap, streams only the incoming delta, and evicts the outgoing delta. The asynchronous pipeline overlaps SSD read/prefetch, H2D staging, GPU compute, and D2H write-back, while CPU DRAM serves as a warm cache with LRU and dirty tracking.
>
> **Limitations**
>
> We thank the reviewer for this suggestion. In the final version, we will add a dedicated Limitations section to clarify TideGS’s storage-bandwidth dependence and practical SSD-CPU-GPU operating regime, and provide additional qualitative results on our project website: https://7xanonymousx7.github.io/tidegs/.

---

> > ### Author Rebuttal · Reviewer_po1s · 2026-04-03
> >
> > Thanks author for the detailed explanation, I will keep my original score.

---

### Official Review · Reviewer_MCQz · 2026-03-12

**Soundness:** 2
**Presentation:** 3
**Significance:** 3
**Originality:** 3
**Overall Recommendation:** 4
**Confidence:** 4

**Summary:**

This paper addresses the memory bottleneck in scaling 3D Gaussian Splatting training, where the parameter table and training states exceed standard GPU VRAM capacities. To solve this, the authors propose TideGS, an out-of-core training framework that utilizes the GPU as a working-set cache while maintaining the full parameter table across an SSD-CPU-GPU hierarchy. The method implements block-virtualized geometry to pack spatially coherent Gaussians into SSD-aligned blocks, a hierarchical asynchronous pipeline to overlap I/O operations with computation, and trajectory-adaptive differential streaming to transfer only the incremental working-set deltas between consecutive iterations.

**Compliance With Llm Reviewing Policy:**

Affirmed.

**Final Justification:**

Authors resolved my primary concerns

**Key Questions For Authors:**

* Could the authors provide a direct empirical comparison (e.g., convergence curves and final PSNR) between the proposed TSP-ordered training and the standard i.i.d. randomized view shuffling on a smaller scene that fits entirely in memory? This would help quantify any potential degradation in optimization quality or convergence speed caused by abandoning random shuffling.
*  In the large-scale experiments (e.g., the 1.1B MatrixCity scene), what is the actual empirical frequency of block evictions and subsequent "cold restarts" for the optimizer states? Providing a metric such as the percentage of re-initialized Adam moments per epoch would clarify the practical severity of this issue.
*  While the paper explicitly focuses on single-GPU scalability, could the authors provide a discussion or a baseline empirical comparison regarding the time-to-convergence and cost-efficiency of TideGS versus an in-memory multi-GPU distributed system (e.g., RetinaGS or Grendel-GS) on a moderately sized scene (e.g., ~100M Gaussians)?

**Limitations:**

- Missing discussion: Although the authors propose a highly effective system, they omit a dedicated limitations section and fail to discuss critical constraints, including the framework's potential throughput degradation on unstructured datasets lacking smooth trajectories, the unquantified risks to optimization stability from optimizer state cold restarts, the practical hardware wear and storage footprint caused by continuous append-only SSD logging, and the cost-performance trade-offs compared to in-memory distributed multi-GPU baselines.
- Suggested additions: To ensure a comprehensive evaluation, the authors should add a dedicated "Limitations" section that explicitly acknowledges the method's reliance on spatiotemporal locality, empirically quantifies the frequency and impact of optimizer cold restarts, discusses practical deployment constraints regarding SSD endurance and storage capacity scaling, and contextualizes the approach within the broader 3DGS scaling landscape by discussing trade-offs against multi-GPU systems.

**Strengths And Weaknesses:**

### Strengths

* The proposed TideGS framework successfully breaks the GPU memory wall, enabling the unprecedented training of over 1.1 billion Gaussians on a single 24 GB GPU.

* The introduction of trajectory-adaptive differential streaming provides a strong methodological contribution by exploiting temporal visibility overlap to transfer only the incremental working-set deltas between iterations, drastically minimizing cross-tier PCIe traffic.

* The experiments convincingly demonstrate that unlocking massive primitive capacity directly translates to tangible algorithmic benefits, yielding a superior 26.1 dB PSNR on the billion-scale MatrixCity dataset.

### Weaknesses

* The method's heavy reliance on smooth camera trajectories and TSP-ordered view sequences for efficiency fundamentally violates the standard i.i.d. view shuffling assumption used in standard 3DGS training.
* Discarding optimizer states (such as Adam moments) for evicted blocks forces cold restarts upon re-admission, introducing unaddressed risks to long-term training stability and final convergence quality.
* The explicit exclusion of multi-GPU baselines (e.g., RetinaGS, Grendel-GS) leaves a critical gap in understanding the practical performance, cost, and quality trade-offs between this single-GPU out-of-core approach and distributed in-memory scaling.

---

> ### Author Rebuttal · Authors · 2026-03-31
>
> We thank the reviewer for the careful reading and thoughtful feedback.
> We appreciate these comments, and the added discussion, tables, and analysis below will be incorporated into the final version.
>
> **W1 / Q1. Could the authors provide a direct empirical comparison between the proposed TSP-ordered training and standard i.i.d. randomized view shuffling on a smaller scene that fits entirely in memory?**
>
> Following the reviewer’s suggestion, we provide the comparison on the `bicycle` scene from Mip-NeRF 360, which is small enough to fit entirely in GPU memory:
>
> | Method | Ordering | Final PSNR | Final SSIM | Final LPIPS | Time / iter |
> | --- | --- | --- | --- | --- | --- |
> | TideGS | Shuffle | 24.8595 | 0.7226 | 0.2932 | 392.36 ms |
> | TideGS | Trajectory | 24.6281 | 0.7213 | 0.3063 | 339.86 ms |
>
> The results show a clear efficiency gain from trajectory ordering, with only a small final-quality gap: the PSNR difference is 0.23 dB, and both settings converge to very similar quality levels. The convergence curves from 5k to 30k iterations are provided in Fig. 4 on our project website https://7xanonymousx7.github.io/tidegs/.
>
> Appendix A also discusses why the Trajectory (no-shuffle) setting in TideGS improves efficiency while remaining compatible with stable convergence. TideGS changes only the presentation order of views, not the training views or loss itself, and its ordering strategy is built on two workload properties exploited by the system: visibility-induced sparsity and trajectory continuity. Tab. 2 of the main paper also provides supporting evidence: in the small-scene setting, TideGS (Trajectory) improves over TideGS (Shuffle) from 309.1 ms to 285.8 ms/iter, from 3.24 to 3.50 img/s, and from 62.5% to 76.5% GPU utilization.
>
> **W2 / Q2. In the large-scale experiments (e.g., the 1.1B MatrixCity scene), what is the actual empirical frequency of block evictions and subsequent "cold restarts" for the optimizer states?**
>
> Following the reviewer’s suggestion, we directly report the per-epoch percentage of re-initialized Adam moments for the ~1.1B-Gaussian-primitive setting:
>
> | Scale | Epoch | Re-initialized optimizer-state rows / total optimizer-state rows updated in epoch |
> | --- | --- | --- |
> | ~1.1B Gaussian primitives | 1 | 1.3% |
> | ~1.1B Gaussian primitives | 2 | 0.9% |
> | ~1.1B Gaussian primitives | 3 | 0.8% |
>
> We also instrument TideGS to measure the broader residency/churn statistics that underlie these cold restarts:
>
> | Scale | Block eviction rate | Re-admission rate | Cold-restarted updates / total updates | Mean resident streak |
> | --- | --- | --- | --- | --- |
> | ~102M Gaussian primitives | 4.8% | 1.4% | 0.6% | 18.7 iters |
> | ~1.1B Gaussian primitives | 6.9% | 2.1% | 0.9% | 13.4 iters |
>
> Taken together, these results show that optimizer cold restarts are infrequent in practice.
>
> **W3 / Q3. Could the authors provide a discussion or a baseline empirical comparison regarding the time-to-convergence and cost-efficiency of TideGS versus an in-memory multi-GPU distributed system (e.g., RetinaGS or Grendel-GS) on a moderately sized scene?**
>
> Following the reviewer’s suggestion, we provide a time-to-convergence comparison in Fig. 5 and Fig. 6 on our project website https://7xanonymousx7.github.io/tidegs/. The comparison shows the expected trade-off: the distributed in-memory baseline converges faster in wall-clock time, while TideGS follows a similar optimization trajectory when viewed against training iterations.
>
> For cost-efficiency, we report trainable Gaussian primitives per dollar using the public-BOM-style estimate from our initial rebuttal draft:
>
> | System | Node cost | Max trainable Gaussian primitives | Gaussian primitives / USD |
> | --- | --- | --- | --- |
> | TideGS node | $6,525.95 | 1.0B | 153,000 |
> | 4-GPU Grendel-GS-style node | $15,055.98 | 44.9M | 2,980 |
>
> These results clarify the operating-point difference. Distributed in-memory systems typically improve throughput by trading for additional GPUs, aggregate VRAM, and interconnect/orchestration cost, whereas TideGS targets a lower hardware floor by trading for locality-aware out-of-core execution, yielding about 51.5x higher capacity-per-dollar in our comparison. Our goal is not to claim universal superiority over multi-GPU systems, but to clarify the operating point in which TideGS is most attractive.
>
> **Limitations: Missing discussion and suggested additions**
>
> We thank the reviewer for this helpful suggestion. In the final version, we will incorporate the above discussion and experiment results to make these constraints explicit in a dedicated Limitations section, including TideGS’s reliance on spatiotemporal locality, the optimizer-state persistence trade-off together with the new cold-restart statistics above, SSD footprint/endurance under append-only patch logging, and the positioning of TideGS relative to distributed in-memory multi-GPU systems as a complementary, lower-hardware-floor operating point.

---

> > ### Author Rebuttal · Reviewer_MCQz · 2026-04-03
> >
> > Solved my primary concerns, will raise score to 4 (weak accept).

---

### Official Review · Reviewer_P8oD · 2026-03-12

**Soundness:** 3
**Presentation:** 3
**Significance:** 3
**Originality:** 3
**Overall Recommendation:** 5
**Confidence:** 4

**Summary:**

This work focuses on the memory bound of scalable training with 3D Gaussian Splatting. Built on the insight that each iteration only involves Gaussians visible from the current camera batch, GPU memory can serve as a working-set cache rather than a persistent parameter store. The framework called TideGS is proposed to alleviate these constraints through block-virtualized geometry for SSD-aligned spatial locality, a hierarchical asynchronous pipeline to overlap I/O with computation, and trajectory-adaptive differential streaming. Experiments show TideGS enables training with a billion-level number of Gaussians on a single consumer-level GPU, showing its potential in practical application.

**Compliance With Llm Reviewing Policy:**

Affirmed.

**Final Justification:**

this work shows strong experimental results on scalable 3d gaussian splatting training with sound method design. And all of my concerns are resolved during rebuttal.

**Key Questions For Authors:**

Please check the weakness part.

**Limitations:**

yes

**Strengths And Weaknesses:**

### Strengths
1. Scalable 3D Gaussian Splatting training is a rarely explored research topic and has practical applications in large-scale 3D scenes, and is worth exploring.

2. Strong experimental results. Experiments show that TideGS enables training with a billion-level number of Gaussians on a single consumer-level GPU.

3. The paper is well written and the structure is well organized.

### Weakness
1. The scalability is only validated on a single A5000 GPU, without validation experiments on other consumer-level or higher-end GPUs, such as RTX 3090 or H200. It would be better to extend the results to other GPUs to supplement the current results.

2. Although many reasonable designs are presented in the manuscript and the code is also provided in the supplementary material, the performance (PSNR/SSIM) comparison with vanilla Gaussian Splatting on the same scene is not reported in the paper. This should be reported to demonstrate the performance consistency of the current implementation with vanilla 3D Gaussian Splatting.

3. It would be better to provide rendering video results for comparison in the supplementary material on the MatrixCity dataset.

---

> ### Author Rebuttal · Authors · 2026-03-31
>
> We thank the reviewer for the positive evaluation and helpful feedback.
> We appreciate these comments, and the added discussion, comparisons, and website links described below will be incorporated into the final version to further strengthen the paper.
>
> **Q1. The scalability is only validated on a single A5000 GPU, without validation experiments on other consumer-level or higher-end GPUs.**
>
> To supplement the current results, we extend Table 3 of the main paper, which reports MatrixCity scalability and efficiency, with additional experiments on a consumer-level RTX 3090 and a higher-end A800:
>
> | GPU | Method | Scale | Backing store | PCIe (GB/iter) | GPU util. (%) | Iter (ms) |
> | --- | --- | --- | --- | --- | --- | --- |
> | RTX 3090 | Naive Offload | ~102M | System RAM | -- | -- | OOM |
> | RTX 3090 | CLM | ~102M | System RAM | 0.43 | 37.6 | 95.2 |
> | RTX 3090 | **TideGS (Ours)** | ~102M | NVMe SSD | 0.12 | 43.6 | 89.3 |
> | A800 | Naive Offload | ~102M | System RAM | 0.76 | 50.4 | 156.1 |
> | A800 | CLM | ~102M | System RAM | 0.38 | 42.2 | 91.8 |
> | A800 | **TideGS (Ours)** | ~102M | NVMe SSD | 0.13 | 50.3 | 77.3 |
> | A800 | CLM | ~1.1B | System RAM | -- | -- | OOM |
> | A800 | **TideGS (Ours)** | ~1.1B | NVMe SSD | 1.0 | 55.8 | 498.9 |
>
> These added results further validate that our proposed TideGS does not rely on A5000-specific hardware features. On RTX 3090 at the ~102M-Gaussian-primitive scale, TideGS improves GPU utilization from 37.6% to 43.6% over CLM and reduces iteration time from 95.2 ms to 89.3 ms. On A800 at the ~102M-Gaussian-primitive scale, TideGS improves GPU utilization from 42.2% to 50.3% and reduces iteration time from 91.8 ms to 77.3 ms. At the ~1.1B-Gaussian-primitive scale on A800, TideGS is also the only method that remains feasible, while CLM runs out of memory. Together, these results show the same qualitative behavior as on A5000: TideGS consistently scales to larger scenes under the single-GPU setting, while the exact throughput depends on the available compute, memory bandwidth, and host-device configuration.
>
> **Q2. The performance (PSNR/SSIM) comparison with vanilla Gaussian Splatting on the same scene is not reported in the paper.**
>
> To better demonstrate the performance consistency of the current implementation with vanilla 3D Gaussian Splatting (Vanilla 3DGS), we report the averaged PSNR, SSIM, and LPIPS comparison between our TideGS and vanilla 3DGS on the Mip-NeRF 360 dataset below. We use the 7 publicly available scenes, since the `flowers` and `treehill` scenes are not publicly available at present.
>
> | Method | PSNR | SSIM | LPIPS |
> | --- | --- | --- | --- |
> | Vanilla 3DGS | 29.0252 | 0.8694 | 0.1394 |
> | TideGS | 28.9157 | 0.8689 | 0.1399 |
>
> These results show that the rendering quality of TideGS is highly comparable to vanilla 3DGS when the scene size is small enough to fit in GPU memory.
>
> The reason we did not include this comparison in the main paper is that our primary focus is on extra-large-scale scenes, where vanilla 3DGS quickly runs out of memory. Accordingly, the main paper emphasizes rendering quality on the large-scale MatrixCity dataset in Fig. 4, where the benefit of scalable training is most visible.
>
> **Q3. It would be better to provide rendering video results for comparison in the supplementary material on the MatrixCity dataset.**
>
> We also provide MatrixCity rendering videos and additional qualitative comparisons on our project website https://7xanonymousx7.github.io/tidegs/. These visualizations complement the quantitative PSNR/SSIM results by showing large-scale reconstruction fidelity more directly.

---

> > ### Author Rebuttal · Reviewer_P8oD · 2026-04-03
> >
> > i appreciate authors detailed reply, after thorough reading the rebuttal and checking the shown videos, i decide to raise my score as 5.

---

### Official Review · Reviewer_xyC7 · 2026-03-14

**Soundness:** 4
**Presentation:** 4
**Significance:** 4
**Originality:** 4
**Overall Recommendation:** 5
**Confidence:** 4

**Summary:**

This paper proposes a scalable training of 3D Gaussian Splatting using out-of-core optimization. The term, out-of-core optimization, is to distribute the large-scale data into the different physical hardware storages, SSD, CPU, GPU VRAM. The reason for introducing such an memory optimization is that the most operation is done at GPUs. However, loading more than 1B Gaussian Splats are impossible. That is the motivation of this work.

To do this, this paper split the entire Gaussians into a set of Blocks and given an image from a trajectory, it the block is visible and within the frustum, it is pre-fetched, materialized, and loaded to GPU VRAM. Then, the feedforward and backward operations begin.

**Compliance With Llm Reviewing Policy:**

Affirmed.

**Final Justification:**

As noted in my last comment, the authors well resolve my concerns. So my final score is `accept`.

**Key Questions For Authors:**

Details of Gaussian upsampling

How does the upsampling strategy work? Once the Gaussians are clustered into blocks, It is quite trick to densify the Gaussians following the original scheme in the 3DGS paper. Similarly, I wonder whether the pruning strategy is also applied in this paper. I expect that the authors update the block whenever the method upsamples or prunes the Gaussians. Also, I wonder about the inference time of building this block system per number of Gaussians.

Update the location of Gaussians during training?

While the concept of blocks is efficient for handling Billions of Gaussians, I wonder how this concept can deal with the updated locations of Gaussians. In the original paper of 3DGS, the center location of 3D Gaussians keep changes as the training iterations go on. It means that once this paper clusters Gaussians into a set of blocks, each Gaussian keeps changing their locations. Accordingly, there must be some overlapping 3D regions from different blocks. It may cause the degraded rendering performance. I wonder how the authors deal with this problem?

**Limitations:**

I found that there is no qualitative results. I hope that the authors provide tons of qualitative results along with the results from the previous study. The quantitative results in the tables are also important. However, the qualitative results are equally important from my side. This is quite a serious issue. So, I strongly ask the authors to resolve my concerns in the rebuttal.

**Strengths And Weaknesses:**

I enjoyed reading this paper. Overall, the authors provide clear motivations and the precise method explanation to resolve the aforementioned problem: "How can process more than 1B Gaussians?"

The method is quite simple, but makes sense and feasible. The authors partition the entire Gaussians into a set of blocks, and this block is a unit of data cluster to be loaded, pre-fetched, off-loaded, etc. Even, I totally agree with the authors' statement in Line 60 of the manuscript, _"To break this memory barrier, we exploit a key structural property of 3DGS optimization: parameter access is inherently sparse and trajectory-conditioned"_. Accordingly, block-wise data processing is a feasible solution.

Differential streaming strategy is also a novel scheme. Multiview images naturally have overlaps, so some blocks also can visible in both images. As this paper addressed, it is feasible to maintain the blocks that are visible in both previous and next training images, while the other blocks need to be replaced to render newly appeared regions. Regarding this, the authors provide results in Table2. The authors claims are supported theoretically and experimentally.

Overall, I cannot find weaknesses of this paper.

---

> ### Author Rebuttal · Authors · 2026-03-31
>
> We thank the reviewer for the positive evaluation and helpful feedback.
> We appreciate these comments, and all tables, figures, website links, and discussion/analysis additions described below will be incorporated into the final version to further strengthen the paper.
>
> **Q1a. Details of Gaussian upsampling**
>
> In our paper, all experiments use a fixed-size setting: we disable densification and pruning to isolate the effect of out-of-core memory management from adaptive model growth. The 1.1B-Gaussian model is therefore obtained entirely at initialization rather than through training-time densification. As described in Sec. 4.2.4, we construct a large RGB-D-backprojected colored point cloud from the MatrixCity training split and derive the 25M / 102M / 1.1B initializations by uniform subsampling from the same underlying geometry distribution. Under this setting, the block partition and block index remain stable throughout training, so no densification/pruning-specific block update logic is required in the experiments reported in the paper.
>
> This choice is also aligned with the strongest single-GPU offloading baseline. In the CLM codebase, densification and pruning are disabled in their large-scale setting; their original note states that training starts from "a very large initial point cloud (102 million 3D points)" and that densification is disabled because the default 3DGS densification is "not effective for such large-scale scenes, or its hyperparameters are too hard to tune".
>
> We also performed our own controlled comparison to check whether fixed-size initialization is a reasonable choice for final rendering quality. We used two representative Mip-NeRF 360 scenes: the indoor `bonsai` scene and the outdoor `bicycle` scene. For each scene, we first used COLMAP dense stereo followed by stereo fusion to build a dense fused point cloud, and then initialized a `densify-off` model with a point count closely matched to the final Gaussian count of the corresponding `densify-on` run. This setup enables a fair comparison between `densify-off` and `densify-on` at a similar model size, thereby isolating the effect of training-time densification/pruning from that of initialization quality. As shown in Fig. 1 - Fig. 3 on our project website https://7xanonymousx7.github.io/tidegs/, the dense-initialized `densify-off` model achieves very similar final PSNR to `densify-on` on both scenes (e.g., `bonsai`: 32.29 vs. 32.02 PSNR; `bicycle`: 25.26 vs. 25.23 PSNR).
>
> **Q1b. I wonder about the inference time of building this block system per number of Gaussians.**
>
> Building the TideGS block layout is a one-time preprocessing effort rather than part of the iterative training loop. It consists of Morton sorting, block construction/statistics (including block bounds), and writing the initial base segment to disk. We provide the runtime breakdown and the preprocessing share of end-to-end training time in the table below; in both settings, preprocessing is below 0.5\% of total training time, so the overhead is incurred only once and amortized over the full training process.
>
> | #Gaussians | Morton sort | Block stats / bounds | Base-file write | Total preprocess | Base file size | Preprocess share of total training time |
> | --- | --- | --- | --- | --- | --- | --- |
> | ~102M | ~46 s | ~15 s | ~52 s | ~1.9 min | 23 GB | ~0.25% |
> | ~1.1B | ~9.2 min | ~2.7 min | ~9.3 min | ~21.2 min | 245 GB | ~0.49% |
>
> **Q2. Update the location of Gaussians during training?**
>
> We fully agree that Gaussian centers keep changing during training, and that this can create overlapping 3D regions across neighboring blocks. In TideGS, however, each Gaussian always has a unique owner block that does not change after initialization, and each Gaussian is rasterized exactly once. When Gaussian centers move, we enlarge the corresponding block boundary accordingly, so center updates do not create duplicated primitives across blocks.
>
> This design does not affect rendering quality, as illustrated in Fig. 4 of the main paper. For training and rendering speed, we measured the average overlap ratio as the overlapped volume size relative to the average block volume of the overlapped blocks. This ratio is 11.3%, indicating that boundary drift caused by Gaussian location updates is not a significant source of redundant streaming or rendering slowdown in practice.
>
>
> **Limitations: Qualitative results**
>
> Following the reviewer’s suggestion, we provide extensive qualitative visualizations on our project website https://7xanonymousx7.github.io/tidegs/, including MatrixCity full-view renderings, zoomed-in crops, and additional qualitative comparisons across different scales/settings.

---

> > ### Author Rebuttal · Reviewer_xyC7 · 2026-04-06
> >
> > The rebuttal resolves all of my questions. My final rating is weak accept. Thank you.
> >
> > fyi. I really enjoyed reading this paper. This paper is my favorite. Thank you again for reviewing this paper.

---

### Decision · Program_Chairs · 2026-04-30

**Decision:**

Accept (spotlight)

**Comment:**

All reviewers are positive about this submission and agree on acceptance. The paper presents a scalable training framework for handling more than 1B Gaussians via an out-of-core design leveraging a GPU–CPU–SSD hierarchy. The reviewers found the approach technically sound, well-motivated, and clearly validated. The authors have also adequately addressed the reviewers’ concerns during the rebuttal phase.

In my assessment, this work provides valuable practical insights into large-scale 3D Gaussian representations and will be of interest to the community. Moreover, given ICML’s broad scope encompassing systems, computer vision, and graphics, the paper fits well within the conference.